# MindLLM: A Subject-Agnostic and Versatile Model for fMRI-to-Text Decoding

Weikang Qiu [1]  Zheng Huang [* 2]  Haoyu Hu [* 3]  Aosong Feng [1]  Yujun Yan [2]  Rex Ying [1]

## Abstract

Decoding functional magnetic resonance imaging (fMRI) signals into text has been a key challenge in the neuroscience community, with the potential to advance brain-computer interfaces and uncover deeper insights into brain mechanisms. However, existing approaches often struggle with suboptimal predictive performance, limited task variety, and poor generalization across subjects. In response to this, we propose MindLLM, a model designed for subject-agnostic and versatile fMRI-to-text decoding. MindLLM consists of an fMRI encoder and an off-the-shelf LLM. The fMRI encoder employs a neuroscience-informed attention mechanism, which is capable of accommodating subjects with varying input shapes and thus achieves high-performance subject-agnostic decoding. Moreover, we introduce Brain Instruction Tuning (BIT), a novel approach that enhances the model's ability to capture diverse semantic representations from fMRI signals, facilitating more versatile decoding. We evaluate MindLLM on comprehensive fMRI-to-text benchmarks. Results demonstrate that our model outperforms the baselines, improving downstream tasks by 12.0%, unseen subject generalization by 24.5%, and novel task adaptation by 25.0%. Furthermore, the attention patterns in MindLLM provide interpretable insights into its decision-making process. Code is available at https://github.com/Graph-and-Geometric-Learning/MindLLM.

## 1. Introduction

Decoding human brain activity (fMRI) to text has sparked significant interest within the neuroscience community (Xia et al., 2024; Chen et al., 2023a; Luo et al., 2023;

---

[*]Equal contribution  [1]Yale University [2]Dartmouth College [3]University of Cambridge. Correspondence to: Weikang Qiu <weikang.qiu@yale.edu>.

*Proceedings of the 42nd International Conference on Machine Learning*, Vancouver, Canada. PMLR 267, 2025. Copyright 2025 by the author(s).

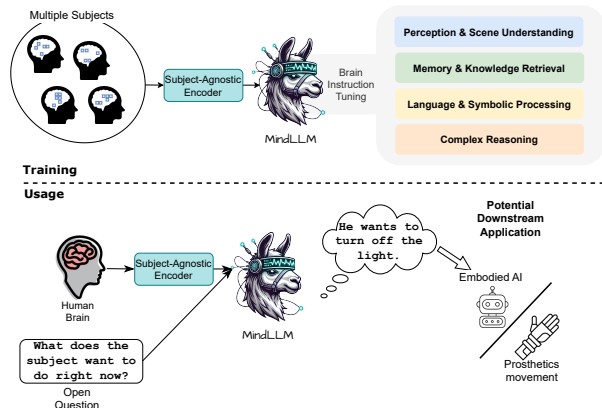

*Figure 1.* The overview of our method. MindLLM is equipped with a subject-agnostic fMRI encoder and an off-the-shelf LLM. MindLLM is trained on multiple subjects with varying input shapes and an instruction-tuning dataset, aiming to encode different facets of semantic information in fMRI. After training, MindLLM is capable of various text decoding tasks. One application is that the decoded contents can be used to achieve neural control of existing systems that are not designed for it.

Hmamouche et al., 2024). The ability to translate brain activity patterns into natural language carries both academic and societal importance. For neuroscientists, it provides deeper and novel insights into cognition, behavior, and consciousness (Qiu et al., 2023; Luo et al., 2023). On a societal level, it presents opportunities for medical applications and improves human-computer interaction (HCI) (Bernal et al., 2022; Du et al., 2022). For example, for individuals with speech impairments, this technology could restore communication capabilities, enabling them to express their thoughts effortlessly (Card et al., 2024). Moreover, as shown in Figure 1, it benefits healthy individuals by allowing neural control of digital devices, such as embodied AIs or prosthetic limbs, allowing for more intuitive and precise movements.

Despite its potential, decoding brain activity to language still faces significant challenges. One major obstacle is the need for versatile decoding tailored to specific applications. For example, decoding may aim to translate a subject's movement intention to control a prosthesis, or to interpret abstract thoughts or memories. Traditional models fail to accommodate such diverse requirements. To address this, UMBRAE (Xia et al., 2024) integrates a Visual Language Model (VLM) (Chen et al., 2023b) and learns to map from

fMRI data to corresponding stimulus images. While this approach achieves versatility to some extent, it remains constrained to tasks directly tied to the current stimulus image and cannot address broader tasks, such as retrieving memories of past visual experiences.

Another critical challenge lies in designing a unified and subject-agnostic architecture. Current methods of brain multimodal decoding mostly rely on a preprocessing step: selecting responsive voxels by comparing the task-based fMRI to the resting-state fMRI. The selection typically results in higher performance compared to using whole-brain data. However, the varying number and irregular spatial distribution of selected voxels across subjects pose significant challenges for developing a unified architecture. To this end, recent studies (Wang et al., 2024a;b) have proposed pooling or sampling voxels to standardize input dimensions. However, as illustrated in Figure 3, these methods still suffer from the loss of spatial information and uneven representations of certain areas, ultimately compromising performance.

**Present Work** Here we present MindLLM, a subject-agnostic and versatile model for fMRI-to-text decoding. Our approach consists of a subject-agnostic fMRI encoder and an off-the-shelf LLM. The subject-agnostic fMRI encoder incorporates a neuroscience-informed attention layer with learnable *queries*, enabling dynamic feature extraction by leveraging both spatial information and neuroscientific priors of voxels, thereby significantly enhancing prediction accuracy. The design of *values* and *keys* separates the voxel's functional information-which is largely consistent across individuals-from its fMRI value, allowing the model to benefit from shared priors across subjects and enhancing generalization to novel subjects. Moreover, to address the challenge of versatile decoding, we propose Brain Instruction Tuning (BIT). BIT trains the model using a diverse dataset that employs images as intermediaries, encompassing tasks designed to capture diverse aspects of semantic information encoded in fMRI data, including perception & scene understanding, memory & knowledge retrieval, language & symbolic processing, and complex reasoning. Figure 1 illustrates the corresponding components.

We evaluate our model on comprehensive benchmarks. Results reveal it outperforms baselines with 12.0% average improvement in various downstream tasks and 24.5% improvement in generalization on unseen subjects. Additionally, we show that our model adapts effectively to novel tasks, demonstrating high customizability and flexibility in real-world applications. Furthermore, our analysis of attention weights offers valuable insights into the working mechanism of our fMRI encoder.

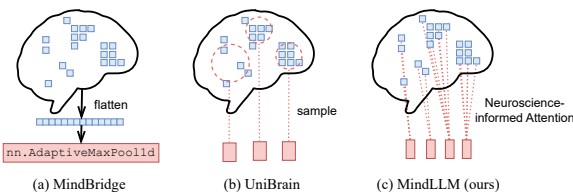

Figure 2. Comparison between our model and previous unified models. MindBridge (Wang et al., 2024a) flattens the voxels and adaptively pools them to a fixed dimension, which overlooks the rich information in positions. UniBrain (Wang et al., 2024b) uniformly samples a subset of voxels and aggregates their neighbors. Different from these methods, we propose neuroscience-informed attention, where each query token attends to all voxels, which minimizes potential information loss in pooling or sampling.

## 2. Related Works

### Brain-Conditioned Text Generation

This line of research mostly focuses on decoding perceived visual stimuli into natural language from fMRI signals. MindGPT (Chen et al., 2023a), UniBrain(Mai & Zhang, 2023) and BrainCap (Ferrante et al., 2023) employ an fMRI encoder guided by CLIP (Radford et al., 2021) and use a language model (Radford et al., 2019; Wang et al., 2022) to decode natural language from the encoded representations. BrainChat (Huang, 2024) utilizes multiple pretraining strategies (Devlin, 2018; He et al., 2022; Yu et al., 2022) to align fMRI with image and text embeddings. These methods fall short in performance and versatility. UMBRAE (Xia et al., 2024) proposes to learn a mapping from fMRI to stimulus images, which later serves as a proxy input for an off-the-shelf visual language model (VLM). Although they achieve performance improvements, the strategy prevents the model from performing tasks that are not directly related to the stimulus images (e.g., answering memory-related questions). In contrast, our end-to-end Brain Instruction Tuning (BIT) ensures seamless and versatile fMRI-to-text decoding, offering the potential to tackle tasks beyond vision-related ones.

**Cross-subjects Decoding** In voxel-level machine learning for brain decoding, the number of voxels varies between subjects (Allen et al., 2022). Most prior works (Scotti et al., 2024a;b) use an MLP for each subject individually. However, due to the fixed input size required by MLP architectures, these models cannot handle varying input shapes. As illustrated in Figure 2, MindBridge (Wang et al., 2024a) proposed to use an adaptive max pooling layer to standardize the input shapes. However, unlike images, which are considered translation invariance, positions in fMRI carry specific bio-informative significance that pooling operations may overlook. UniBrain (Wang et al., 2024b) proposed to sample groups of voxels. Such a sampling strategy, on the one hand, may lead to information loss if some voxels are

not included in any group. On the other hand, the irregular spatial distribution of 3D voxels with varying density and curvature may result in underrepresentation or overrepresentation of certain areas. Different from these methods, our model employs a neuroscience-informed attention mechanism that accounts for every single voxel while preserving their bio-informative positional information, ensuring a more comprehensive and precise representation.

**Multi-Modal Large Language Model** Aiming to augment the perceptual capacities of Large Language Models (LLMs), there has been a growing interest in extending them to handle multiple modalities within a unified model. Numerous studies have attempted to incorporate modalities such as images (Alayrac et al., 2022; Zhang et al., 2023b; Wang et al., 2023b), videos (Cheng et al., 2024; Kondratyuk et al., 2023; Zhang et al., 2023a), and point clouds (Xu et al., 2023; Qi et al., 2025). OneLLM (Han et al., 2024a) stands out by aligning eight different modalities, including fMRI, with language. However, their approach employs an individual convolution network for each subject instead of a unified architecture for fMRI encoding across subjects, which restricts its applicability to new subjects in real-world scenarios. Furthermore, the approach solely relies on captions as textual annotations, which limits the model's capability for versatile fMRI decoding.

**Multimodal fMRI Decoding** Beyond text, recent studies have explored decoding brain signals into other modalities, including images (Scotti et al., 2024b; Wang et al., 2024a; Chen et al., 2023c), videos (Sun et al., 2024; Chen et al., 2023d), speech (Chen et al., 2024) and music (Denk et al., 2023), as well as in reverse directions (Toneva & Wehbe, 2019). The encoder architectures in these works typically either adopt transformers for time-series data, which is distinct from our voxel-based setting, or fall into one of the architectures we have discussed in previous paragraphs. Therefore, focusing on text generation, our work improves the fMRI encoder architecture for voxel-level inputs, and designs novel objectives (i.e., brain instruction tuning) for our target modality.

## 3. Method

In this section, we propose a neuroscience-informed fMRI encoder designed to achieve high-performance, subject-agnostic decoding. To further enable versatile decoding, we introduce the construction of a brain instruction tuning dataset, which captures diverse semantic representations encoded in fMRI data.

### 3.1. Method Overview

As illustrated in Figure 3, our model consists of an fMRI encoder $f_\theta$ and an off-the-shelf LLM. In practice, we use

Vicuna-7b (Zheng et al., 2023) as our LLM to maintain consistency with our baseline (Xia et al., 2024). For each sample, let $\boldsymbol{v} = [v_1, v_2, \cdots, v_N] \in \mathbb{R}^N$ be the fMRI signals of input voxels, where $N$ is the number of voxels. Note that $N$ varies between different subjects, ranging from $12,682$ to $17,907$ in the dataset we use (Allen et al., 2022).

The fMRI encoder $f_\theta$, featuring a neuroscience-informed attention layer, encodes $\boldsymbol{v}$ to fMRI tokens $X_v = [\boldsymbol{x}_{v,1}, \boldsymbol{x}_{v,2}, \cdots, \boldsymbol{x}_{v,L}] \in \mathbb{R}^{d \times L}$, where $L$ is the number of tokens and $d$ is the dimension of token embeddings. We then prepend these learned fMRI tokens to the language tokens in the BIT dataset we propose.

### 3.2. fMRI Encoder

As mentioned before, currently most models for fMRI decoding can not handle varying input shapes and are not subject-agnostic, with only a few exceptions (Wang et al., 2024b). However, these exceptions still suffer from information loss and uneven representations of certain brain areas. To this end, we propose a novel neuroscience-informed attention mechanism to accommodate varying voxel numbers across subjects, enabling a subject-agnostic encoding strategy. Below we talk about the design of *queries* $\{\boldsymbol{q}_i\}$, *keys* $\{\boldsymbol{k}_i\}$ and *values* $\{\boldsymbol{v}_i\}$ in the attention layer. For *values*, we directly use the fMRI signal of each voxel, which means $\boldsymbol{v_i} = v_i \in \mathbb{R}$. Making each voxel a *value* token maximally prevents information loss compared to pooling- (Wang et al., 2024a) or sampling-based (Mai & Zhang, 2023) methods. The *queries* are randomly initialized and learnable. We expect each *query* to represent a certain pattern of the brain (refer to visualizations in Section 4.8). The design of *keys* will be discussed below.

**Exclude fMRI values from *keys*** The vanilla cross attention (Zhu et al., 2020; Vaswani, 2017) derives both *keys* and *values* from the same input source. However, we found this would lead to poor performance in fMRI. We argue the reason: different from images or text, which are usually considered translation-invariant, the positions of voxels carry specific brain *functional information*, as voxels in different areas are associated with distinct brain functions. Consequently, a voxel's position alone can theoretically serve as effective *keys* for attention weight computation (McManus, 2002; Zhang et al., 2023c; Liu et al., 2024). Including fMRI values into *keys*, however, introduces additional noise instead of valuable information, thus resulting in poorer performance. Moreover, since brain regions tend to serve similar functions across individuals, decoupling voxel positions from fMRI signals can facilitate the sharing of priors across subjects, potentially improving generalization to unseen subjects.

In light of this, instead of the vanilla cross attention, which derives the *keys* and *values* from the same inputs, we ex-

clude the fMRI value of each voxel and use its positional information alone as its *key* embedding. The positional information is encoded from the coordinates of each voxel, i.e. $\boldsymbol{k}_i^{\text{pos}} = \text{PE}(\boldsymbol{c}_i)$ for the $i$-th voxel, where $\boldsymbol{c}_i \in \mathbb{R}^3$ denotes the coordinates of the voxel. In practice, we use the Fourier positional encoding proposed in (Tancik et al., 2020) due to its superiority in encoding coordinate information.

**Incorporation of Brain Parcellations** While positional encoding alone improves performance, it lacks inherent neuroscientific grounding, potentially making it challenging for the model to efficiently learn representations aligned with established principles of brain function. To overcome this, we incorporate existing brain region parcellations (Glasser et al., 2016; Rolls et al., 2020) into the *key* embeddings. Formally, given a parcellation $\mathcal{P}$, with regions indexed by $1, \cdots, N_{\mathcal{P}}$. Let $\mathcal{P}(i) \in [1, 2, \cdots, N_{\mathcal{P}}]$ be the region that the $i$-th voxel belongs to, and $E[\mathcal{P}(i)] \in \mathbb{R}^d$ be the corresponding learnable embedding of the region, which will be incorporated in the *key* embeddings as $\boldsymbol{k}_i^{\text{reg},\mathcal{P}} = E[\mathcal{P}(i)] \in \mathbb{R}^d$.

**Combining Multiple Parcellations** It is crucial to choose an appropriate brain region parcellation. Previous region-based methods (Qiu et al., 2023; Li et al., 2021; Kan et al., 2022) can usually only choose one arbitrarily. In contrast, our model design allows us to combine multiple parcellations $\mathcal{P}^1, \mathcal{P}^2, \cdots$ by concatenating their respective region encodings to the *key* embeddings. In conclusion, the final *key* embeddings are the concatenation of the positional encoding and multiple region encodings,

$$\boldsymbol{k}_i = \boldsymbol{k}_i^{\text{pos}} \| \boldsymbol{k}_i^{\text{reg},\mathcal{P}^1} \| \boldsymbol{k}_i^{\text{reg},\mathcal{P}^2} \| \cdots \tag{1}$$

where $\|$ denotes the concatenation operation. This process is illustrated in Figure 3's lower right part.

The positional and region encodings complement each other: The region encodings serve as coarse-scale features, providing a neuroscientific-grounded basis, while the fine-scale positional encoding allows our model to learn finer-grained information directly from the data.

This attention design separates a voxel's *functional information*—which is largely consistent across individuals—from its fMRI value, thereby enhancing generalization. Instead of relying on pooling or sampling, the attention mechanism employs learnable aggregation, while the integration of positional encoding and neuroscientifically informed region encodings further ensures high performance.

After the attention layer, we obtain the hidden representations $\boldsymbol{z}_q \in \mathbb{R}^{N_q}$ where $N_q$ is the number of query embeddings. We then employ an MLP and a reshape operation to map the hidden representations to $L$ fMRI tokens, i.e., $X_v = \text{reshape}\left(\text{MLP}(\{\boldsymbol{z}_q\})\right) \in \mathbb{R}^{L \times d}$.

The process of the fMRI encoder is illustrated in Figure 3. The obtained fMRI tokens are then prepended to the lan-

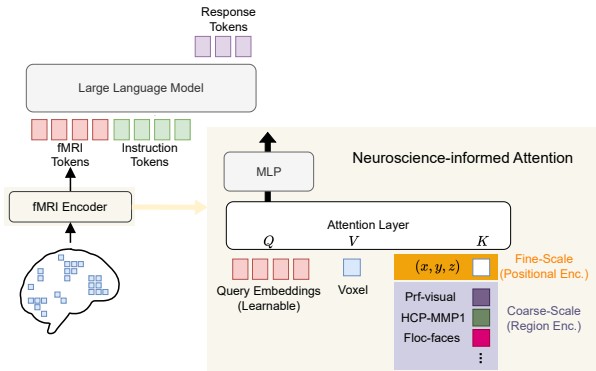

*Figure 3.* Model Architecture. The fMRI encoder maps fMRI to a series of fMRI tokens through our proposed neuroscience-informed attention. The large language model, with both fMRI and text tokens, will be trained by brain instruction tuning.

guage tokens in conversations.

In addition, we provide a side-by-side comparison between our encoder and existing ones in Appendix F.

### 3.3. Brain Instruction Tuning (BIT)

To enable versatile fMRI-to-text decoding, an appropriate BIT dataset is required, yet no such dataset currently exists. To bridge this gap, we construct one based on the fact: MSCOCO images (Chen et al., 2015) serve as stimuli for fMRI recordings in the fMRI study (Allen et al., 2022), and an abundance of datasets provide text annotations (e.g., VQA) for MSCOCO images. Using the images as intermediaries, we select those relevant to brain functions and pair the fMRI data with corresponding text annotations. For example, given an image of a billboard with annotated textual content, we can reasonably infer that when a subject perceives textual information (e.g., contents on the billboard), corresponding representations are encoded in the brain. This suggests the possibility of extracting such information from fMRI signals. We select datasets to fulfill various purposes, enabling the model to capture diverse aspects of semantic information embedded in fMRI signals, including visual perception & scene understanding, language & symbolic processing, memory & knowledge retrieval and complex reasoning, which are considered among most fundamental and essential properties of human brains (Robertson, 2002; Stenning & Van Lambalgen, 2012; Wade & Swanston, 2013; Friederici, 2017).

**Perception & Scene Understanding** As illustrated in Figure 4, we begin by using caption tasks at both coarse and fine-grained levels to train the model's ability to understand and summarize what the subject perceives visually (Chen et al., 2015; Krause et al., 2017). Additionally, we incorporate QA tasks (Ren et al., 2015; Krishna et al., 2017;

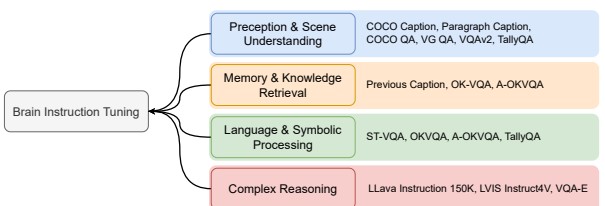

*Figure 4.* Dataset Taxonomy in Brain Instruction Tuning.

[Acharya et al., 2019](#)) to enhance the model's ability to retrieve and reason about visually perceived content.

**Memory & Knowledge Retrieval** To go beyond tasks directly related to present visual perception, we construct the *previous captioning* task, a memory-oriented task that challenges the model to caption images that the subject previously viewed, simulating memory recall processes. Furthermore, we aim to encode knowledge structures in human brains. The OK-VQA ([Marino et al., 2019](#)) and A-OKVQA ([Schwenk et al., 2022](#)) datasets include questions requiring external knowledge that is not present in the image but resides in human brains. For example, A photo of a hydrant may prompt the answer "firetruck," even though the firetruck is absent in the image. This association also reflects the way human cognition operates through a network of interconnected meanings, where one concept unconsciously triggers another. Such a process, which is called "slippage of the signifier" ([Lacan, 2001](#); [1988](#); [Miller & Lacan, 2018](#)), highlights the symbolic processes through which the brain constructs and retrieves meaning.

**Language & Symbolic Processing** In addition to the aforementioned OK-VQA and A-OKVQA datasets, which are also related to symbolic process, we further combine datasets of text recognition ([Biten et al., 2019](#)) and numerical reasoning ([Acharya et al., 2019](#)) to facilitate this aspect.

**Complex Reasoning** Finally, we try to approximate the reasoning process that happens in human brains with datasets ([Liu et al., 2023](#); [Wang et al., 2023a](#); [Li et al., 2018](#)) that require intricate logical and inferential processes. We expect these datasets to challenge the model to extract the reasoning process, drawing upon both visual understanding and abstract problem-solving, thus bridging perception, memory, and knowledge into a cohesive cognitive framework.

We ended up with a brain instruction tuning dataset consisting of $980,610$ conversations associated with fMRI recordings from 15 datasets. Appendix A lists the instructions and other details for each dataset. The instruction tuning enables versatile fMRI-to-text decoding. In particular, the introduction of tasks like *previous caption* empowers the model to perform a broader range of tasks beyond vision-related ones, which the previous model ([Xia et al., 2024](#)) fails.

To train the model with the BIT dataset, for each

sample $\boldsymbol{v}$, we sample a multi-run conversation $X_t = (X_u^1, X_a^1, \cdots, X_u^T, X_a^T)$ from all conversations associated with it, where $T \geq 1$ represents the number of turns. $a$ indicates the message from the assistant and $u$ indicates the message is from the user. The training objective is to maximize the probability of the assistant's response only

$$\arg\max_{\theta} p(X_a|X_v, X_{\text{inst}}) = \prod_{t=1}^{T} p(X_a^t|X_u^{\leq t}, X_a^{\leq t}, X_{\text{inst}}, X_v)$$

Figure 5 illustrates the chat template and the training objective. We freeze the weights of the LLM and only train the fMRI encoder since we want to preserve the LLM's language modeling prior and ensure a fair comparison with baselines such as [Xia et al. (2024)](#).

**Computational Complexity** According to the analysis in Appendix C, our model does not introduce additional complexity compared to previous methods ([Scotti et al., 2024b](#); [Wang et al., 2024a](#)).

```
<system message>
user: X_v, X_inst, X_1^u
assistant: X_1^a
user: X_2^u
assistant: X_2^a
......
```

*Figure 5.* The chat template used during instruction tuning, illustrating two turns of conversations. Two turns of conversation are shown. Tokens highlighted in magenta are used for next-token prediction loss computation.

## 4. Experiments

In this section, we first evaluate our model on various downstream tasks, demonstrating its versatile decoding capabilities. Next, we assess its generalizability to novel subjects and its adaptability to real-world applications. Finally, we analyze the functions of queries in our neuroscience-informed attention mechanism.

### 4.1. Settings

**fMRI Datasets** We use the widely used Natural Scenes Dataset (NSD) ([Allen et al., 2022](#)), a large-scale dataset consisting of fMRI measurements of 8 healthy adult subjects. During data collection, subjects viewed images from the MS-COCO dataset ([Lin et al., 2014](#)) and were instructed to press buttons to indicate whether they had previously seen each image.

**Downstream Datasets** The downstream dataset will be discussed within each experiment section. See examples and a short description for all datasets we will use in Appendix A. Implementation details could be found in Appendix B.

## 4.2. Brain Captioning

To evaluate the model's performance on downstream tasks, we start with the widely used brain captioning benchmark (Xia et al., 2024). The task, built upon COCO Caption (Chen et al., 2015) requires the model to predict captions of given images as fMRI stimuli.

**Baselines** The following baselines are considered in this experiment: SDRecon (Takagi & Nishimoto, 2023), UniBrain (Mai & Zhang, 2023), and BrainCap (Ferrante et al., 2023) employs a linear regression, mapping the fMRI to the inputs of an image caption model (Li et al., 2023). OneLLM (Han et al., 2024a) is a multimodal large language models that align 8 modalities (including fMRI) with language all in one model. For fair and efficient comparison, we only finetune the encoder, given that we freeze the LLM in our method as well. UMBRAE learns an encoder that maps fMRIs to images through an encoder similar to the MLP mixer (Tolstikhin et al., 2021). BrainChat (Huang, 2024) segments the flattened voxels into 16 patches and employs a transformer to decode text conditioned on the patches. It is worth noting that all of these baselines require subject-specific layers or parameters. In contrast, our model is subject-agnostic, thus with the potential to generalize on novel subjects.

**Metric** Following previous works, we use five standard metrics for text generation: BLEU-$k$ (Papineni et al., 2002), ROUGE-L (Lin, 2004), CIDEr (Vedantam et al., 2015), SPICE (Anderson et al., 2016), METEOR (Banerjee & Lavie, 2005).

Table 1 shows that our model outperforms baselines in terms of most metrics, with an average improvement of 3.32%, even if our model does not have any subject-specific layers. We argue that this is attributed to both the novel architecture design and the introduction of BIT, which will be evident in the next experiment.

## 4.3. Versatile Decoding

The purpose of experiments in this section is two-fold: 1) To investigate the impact of our model design and the introduction of BIT on performance improvement. 2) To evaluate

the versatility of the model, i.e., its performance on various downstream tasks.

**Baselines** Besides baselines that could be adapted to this experiment from the previous one, we further consider the following subject-agnostic models as baselines. 1) Mind-Bridge (Wang et al., 2024a) flattens the voxels and adaptively adjusts the padding and stride to pool the voxels into a fixed dimension. The original implementation of Mind-Bridge has subject-specific parameters. However, since those parameters are of the same size, we make them shared across subjects and thus make the model subject-agnostic. 2) UniBrain (Wang et al., 2024b) samples voxels into a fixed number of groups and employs a transformer where groups are treated as tokens. This UniBrain is unrelated to the UniBrain in the previous section; they just share the same name.

**Datasets & Metric** We use the test split of all QA & caption datasets in the BIT dataset. We strictly adhere to the official metrics on all datasets. In summary, for sentence generation, we use BLEU-$k$ (Papineni et al., 2002), ROUGE-L (Lin, 2004), CIDEr (Vedantam et al., 2015), SPICE (Anderson et al., 2016), METEOR (Banerjee & Lavie, 2005). For QA-related tasks, we use VQA accuracy (Antol et al., 2015) as well as special metrics proposed in the original paper (e.g. ANLS for ST-VQA (Biten et al., 2019)).

The results on subject 1 are shown in Table 2. Our model outperforms baselines, with an average improvement of 12.0%. Further, by comparing instruction tuning and from-scratch models, we find that instruction tuning has a significant positive effect, with an average improvement of 28.0%. The results remain stable across different random seeds; for instance, according to our observations, the BLEU-1 score for paragraph captioning exhibits a maximum of ±0.3 variance. We report experiments on other subjects in Appendix G.

## 4.4. Unseen Subject Generalization

Our neuroscience-informed, subject-agnostic design enhances generalization to novel subjects, a crucial factor in real-world applications where training a model for each

*Table 1.* Results of brain captioning. The CIDEr metric is scaled by a factor of 100 for consistency with Table 1 and baselines.

| Method | subject agnostic | BLEU-1 ↑ | BLEU-2 ↑ | BLEU-3 ↑ | BLEU-4 ↑ | METEOR ↑ | ROUGE ↑ | CIDEr ↑ | SPICE ↑ |
|---|---|---|---|---|---|---|---|---|---|
| SDRecon (Takagi & Nishimoto, 2023) | ✗ | 36.21 | 17.11 | 7.22 | 3.43 | 10.03 | 25.13 | 13.83 | 5.02 |
| OneLLM (Han et al., 2024a) | ✗ | 47.04 | 26.97 | 15.49 | 9.51 | 13.55 | 35.05 | 22.99 | 6.26 |
| UniBrain (Mai & Zhang, 2023) | ✗ | – | – | – | – | 16.90 | 22.20 | – | – |
| BrainCap (Ferrante et al., 2023) | ✗ | 55.96 | 36.21 | 22.70 | 14.51 | 16.68 | 40.69 | 41.30 | 9.06 |
| BrainChat (Huang, 2024) | ✗ | 52.30 | 29.20 | 17.10 | 10.70 | 14.30 | 45.70 | 26.10 | – |
| UMBRAE (Xia et al., 2024) | ✗ | 59.44 | 40.48 | 27.66 | 19.03 | 19.45 | 43.71 | **61.06** | **12.79** |
| UniBrain (Wang et al., 2024b) | ✓ | 59.08 | 39.64 | 26.36 | 17.68 | 17.49 | 43.48 | 48.20 | 9.38 |
| MindLLM (Ours) | ✓ | **61.75** | **42.84** | **29.86** | **21.24** | **19.54** | **45.82** | 60.97 | 11.79 |

*Table 2.* Versatile decoding. A dash − means the model could not perform this task. The superscript ° means the model is trained from scratch, in contrast to their BIT version. The CIDEr metric is scaled by a factor of 100 for consistency with Table 1 and baselines.

| | | OneLLM | UMBRAE | BrainChat | MindBridge° | UniBrain° | MindLLM° | MindBridge | UniBrain | MindLLM |
|---|---|---|---|---|---|---|---|---|---|---|
| subj-agnostic | | ✗ | ✗ | ✗ | ✓ | ✓ | ✓ | ✓ | ✓ | ✓ |
| COCO-QA | Accuracy↑ | 11.09% | 22.23% | 39.44% | 40.19% | 38.38% | 42.09% | 45.33% | 42.00% | **48.19%** |
| VG-QA | Accuracy↑ | 8.76% | 19.67% | 21.00% | 20.84% | 21.27% | 21.68% | 23.53% | 24.02% | **24.06%** |
| VQA-v2 | Accuracy ↑ | 33.68% | 51.23% | 40.02% | 43.25% | 46.04% | 44.13% | 47.91% | 48.58% | **52.14%** |
| A-OKVQA | Accuracy ↑ | 25.23% | 43.24% | 20.52% | 22.12% | 19.47% | 29.20% | 50.44% | 43.36% | **52.21%** |
| ST-VQA | ANLS ↑ | 5.74% | 5.46% | 9.58% | 10.20% | 7.01% | 12.76% | 11.64% | 8.76% | **12.92%** |
| OK-VQA | Accuracy ↑ | 22.98% | 10.35% | 17.22% | 27.63% | 18.63% | 27.70% | 32.13% | 32.30% | **33.33%** |
| TallyQA | Accuracy ↑ | 8.34% | 44.10% | 43.22% | 43.49% | 44.83% | 43.75% | 49.46% | 53.77% | **54.76%** |
| | RMSE ↓ | 7.45 | 3.94 | 1.90 | 2.03 | 1.83 | 2.04 | 1.86 | **1.67** | 1.76 |
| Paragraph Caption | BLEU-1↑ | 0.26 | **29.82** | 22.21 | 21.82 | 25.69 | 26.49 | 25.69 | 28.28 | 29.43 |
| | BLEU-2 ↑ | 0.08 | 14.26 | 10.23 | 10.47 | 12.62 | 12.48 | 13.00 | 15.47 | **15.78** |
| | BLEU-3↑ | 0.03 | 6.52 | 6.38 | 5.58 | 6.70 | 6.43 | 7.10 | 8.90 | **9.14** |
| | BLEU-4↑ | 0.01 | 2.95 | 2.12 | 3.14 | 3.81 | 3.63 | 4.22 | **5.60** | 5.51 |
| | METEOR ↑ | 2.36 | 12.60 | 9.10 | 10.95 | 11.13 | 10.71 | 11.39 | **13.50** | 13.18 |
| | CIDEr ↑ | 0.00 | 7.39 | 6.02 | 7.50 | 3.92 | 2.44 | 3.55 | 1.82 | **7.80** |
| VQA-E | Accuracy ↑ | 19.60% | 47.84% | 46.20% | 45.40% | 44.42% | 44.55% | 48.48% | 48.39% | **50.95%** |
| | BLEU-1 ↑ | 17.32 | 29.83 | 35.99 | 35.63 | 35.30 | 35.08 | 36.18 | 37.26 | **37.70** |
| | BLEU-2 ↑ | 7.44 | 14.76 | 18.33 | 18.27 | 18.04 | 17.82 | 19.38 | 20.41 | **20.56** |
| | BLEU-3 ↑ | 3.62 | 8.17 | 10.01 | 10.32 | 10.20 | 10.05 | 11.30 | 12.25 | **12.34** |
| | BLEU-4 ↑ | 1.82 | 4.87 | 6.60 | 6.27 | 6.14 | 6.00 | 7.00 | 7.83 | **7.92** |
| | CIDEr ↑ | 19.32 | 63.26 | 78.33 | 79.05 | 77.31 | 76.80 | 86.62 | 92.09 | **93.60** |
| | METEOR ↑ | 6.69 | 12.25 | 13.64 | 14.13 | 13.89 | 13.96 | 14.81 | 15.51 | **15.62** |
| | ROUGE ↑ | 16.84 | 28.38 | 32.82 | 33.78 | 33.25 | 33.11 | 34.56 | 35.87 | **35.88** |
| FSVQA | VQA Acc. ↑ | 31.44% | 40.67% | 36.30% | 42.00% | 37.05% | 42.53% | 45.95% | 44.58% | **48.03%** |
| | FSVQA Acc. ↑ | 21.02% | 0.00% | 30.22% | 37.40% | 32.30% | 38.50% | 40.97% | 37.87% | **43.00%** |
| | BLEU-1 ↑ | 37.42 | 23.11 | 83.99 | 85.68 | 83.84 | 85.88 | 86.52 | 85.10 | **87.10** |
| | BLEU-2 ↑ | 31.72 | 5.86 | 78.50 | 81.27 | 78.81 | 81.62 | 82.28 | 80.01 | **83.03** |
| | BLEU-3 ↑ | 26.95 | 2.10 | 73.00 | 77.10 | 73.97 | 77.62 | 78.34 | 75.49 | **79.27** |
| | BLEU-4 ↑ | 22.48 | 1.04 | 69.73 | 72.89 | 68.91 | 73.56 | 74.35 | 70.73 | **75.50** |
| | METEOR ↑ | 26.35 | 8.93 | 44.76 | 47.59 | 45.94 | 47.96 | 48.63 | 46.89 | **49.05** |
| | CIDEr ↑ | 312.75 | 4.07 | 600.00 | 636.40 | 609.00 | 646.26 | 657.02 | 628.83 | **666.26** |
| Previous Caption | BLEU-1 ↑ | 41.86 | − | 21.19 | 21.17 | 24.84 | 44.52 | 42.45 | 43.01 | **47.20** |
| | BLEU-2 ↑ | 19.44 | − | 8.00 | 7.57 | 9.70 | 22.46 | 20.04 | 20.03 | **25.16** |
| | BLEU-3 ↑ | 9.25 | − | 1.98 | 2.85 | 3.40 | 10.39 | 9.61 | 9.19 | **12.95** |
| | BLEU-4 ↑ | 3.67 | − | 1.02 | 1.28 | 1.46 | 5.45 | 5.31 | 4.58 | **7.49** |
| | METEOR ↑ | 10.14 | − | 6.55 | 6.46 | 7.20 | 11.00 | 10.83 | 10.81 | **11.96** |
| | ROUGE ↑ | 30.19 | − | 21.23 | 20.88 | 23.04 | 33.20 | 32.38 | 31.99 | **34.58** |
| | CIDEr ↑ | 6.65 | − | 9.21 | 8.83 | 11.73 | 9.39 | 7.89 | 7.53 | **16.02** |
| | SPICE ↑ | 2.49 | − | 2.44 | 2.56 | 2.78 | 3.07 | 2.80 | 2.92 | **3.93** |

individual is impractical. To evaluate it, we perform instruction tuning on 7 out of the 8 subjects in the natural scene dataset (Allen et al., 2022), and evaluate generalization on the held-out subject. Table 3 shows our model outperforms two other subject-agnostic baselines in most cases, with an average improvement of 24.5% compared to the second-best model. We anticipate further performance gains through the application of domain adaptation techniques (Xiao et al., 2023; Ganin & Lempitsky, 2015; Ganin et al., 2016; Gong et al., 2012), which we leave as a direction for future work.

### 4.5. Adapting to New Tasks

It is common that users want to adapt the MindLLM to their own specific use cases. To this end, we aim to assess our model's adaptability to new tasks.

**Dataset & Metrics** We use TDIUC (Kafle & Kanan, 2017), a QA dataset consisting of 12 types of questions, as a benchmark to evaluate the model's various capabilities comprehensively. Additionally, we further select 2 task types-*sentiment understanding* and *utility/affordance* tasks, that are particularly relevant to BCI applications as sub-datasets. The *utility/affordance* task, for instance, enables the model to identify useful objects in a given scene and autonomously decide whether to utilize them. Following their paper, we compute the accuracy of each type and report the arithmetic mean-per-type (A-MPT) and the harmonic mean-per-type (H-MPT). For the 2 selected types, we report the accuracy respectively.

Table 4 shows our model achieves balanced (high harmonic mean) and consistently improved performances with an

*Table 3.* Model generalization, compared with subject-agnostic model. We train the models on subject $1-7$ and evaluate on subject 8, which is the held-out subject.

| | | MindBridge | UniBrain | MindLLM |
|---|---|---|---|---|
| COCO-QA | Accuracy ↑ | 35.88 | 24.95 | **38.96** |
| VG-QA | Accuracy ↑ | **20.56** | 16.23 | 19.29 |
| VQA-v2 | Accuracy ↑ | 42.80 | 40.16 | **46.04** |
| A-OKVQA | Accuracy ↑ | 44.55 | 28.71 | **45.54** |
| ST-VQA | ANLS ↑ | 9.33 | 9.30 | **13.63** |
| OK-VQA | Accuracy ↑ | 21.94 | 17.09 | **25.44** |
| TallyQA | Accuracy ↑ | 38.92 | 32.51 | **42.29** |
| | RMSE ↓ | 2.12 | **2.02** | 2.11 |
| COCO-Caption | BLEU-1 ↑ | 39.84 | 41.90 | **44.60** |
| | BLEU-2 ↑ | 19.55 | 19.67 | **24.04** |
| | BLEU-3 ↑ | 9.29 | 8.89 | **12.79** |
| | BLEU-4 ↑ | 5.24 | 4.33 | **7.52** |
| | METEOR ↑ | 10.39 | 10.80 | **11.16** |
| | ROUGE ↑ | 31.10 | 31.54 | **33.47** |
| | CIDEr ↑ | 8.70 | 6.40 | **13.22** |
| | SPICE ↑ | 2.67 | 2.39 | **3.82** |
| Paragraph Caption | BLEU-1 ↑ | 23.18 | 21.73 | **26.07** |
| | BLEU-2 ↑ | 10.71 | 8.94 | **12.64** |
| | BLEU-3 ↑ | 4.61 | 3.72 | **6.27** |
| | BLEU-4 ↑ | 2.22 | 1.92 | **3.46** |
| | METEOR ↑ | 9.99 | 9.47 | **11.18** |
| | CIDEr ↑ | 0.71 | 1.56 | **5.61** |
| VQA-E | Accuracy ↑ | 41.78 | 38.53 | **46.19** |
| | BLEU-1 ↑ | 32.54 | 32.86 | **34.54** |
| | BLEU-2 ↑ | 16.13 | 15.48 | **17.54** |
| | BLEU-3 ↑ | 8.82 | 7.98 | **9.84** |
| | BLEU-4 ↑ | 5.16 | 4.42 | **5.81** |
| | CIDEr ↑ | 68.13 | 58.79 | **73.27** |
| | METEOR ↑ | 12.74 | 12.26 | **13.56** |
| | ROUGE ↑ | 30.63 | 29.38 | **32.36** |
| FSVQA | VQA Acc. ↑ | 42.33 | 37.92 | **43.92** |
| | FSVQA Acc. ↑ | 37.16 | 30.83 | **39.07** |
| | BLEU-1 ↑ | 75.81 | 82.94 | **86.19** |
| | BLEU-2 ↑ | 71.03 | 77.24 | **81.83** |
| | BLEU-3 ↑ | 66.40 | 71.94 | **77.77** |
| | BLEU-4 ↑ | 61.55 | 66.54 | **73.58** |
| | METEOR ↑ | 45.84 | 45.24 | **47.94** |
| | CIDEr ↑ | 428.39 | 587.78 | **646.30** |

average of 13.5%. We could also observe the performance benefits from BIT, with 25.0% absolute improvement.

## 4.6. Performance scale with the number of subjects

We conducted experiments to evaluate how model performance scales with the number of subjects and report performances on the COCO caption task. We examined both the in-distribution (seen subjects) setting in Figure 5 and the out-of-distribution (held-out subjects) setting in Figure 6. Our results show significant performance improvements as the number of training subjects increases, demonstrating that the model benefits from exposure to more subjects during pre-training.

## 4.7. Ablation Study

We conduct ablation studies on the design of key embeddings in the neuroscience-informed attention module in Fig-

*Table 4.* Model adaptation to new tasks. *sentiment understanding* and *utility/affordance* are sub-datasets from TDIUC that are particularly relevant to BCI applications.

| Method | Overall | | Sentiment Understanding | Utility/Affordance |
|---|---|---|---|---|
| | A-MPT | H-MPT | Accuracy | Accuracy |
| MindLLM° | 41.09% | 19.38% | 70.00% | 0.00% |
| MindBridge | 49.77% | 39.88% | 80.00% | 14.29% |
| UniBrain | 51.50% | 36.76% | 80.00% | 28.57% |
| MindLLM | **54.08%** | **45.43%** | **80.77%** | **50.00%** |

*Table 5.* Performance scaling with number of subjects (in-distribution). Evaluation is performed on Subject 1, who is included during training.

| | COCO Caption | | | | | | | |
|---|---|---|---|---|---|---|---|---|
| number of subjects | BLEU-1 ↑ | BLEU-2 ↑ | BLEU-3 ↑ | BLEU-4 ↑ | METEOR ↑ | ROUGE ↑ | CIDEr ↑ | SPICE ↑ |
| 1 | 58.05 | 37.95 | 24.40 | 16.14 | 16.62 | 42.03 | 43.04 | 8.81 |
| 3 | 56.81 | 37.16 | 24.12 | 16.30 | 16.44 | 41.93 | 40.94 | 9.12 |
| 5 | 58.17 | 38.65 | 25.49 | 17.30 | 16.97 | 42.36 | 46.54 | 9.19 |
| 7 | 58.52 | 39.20 | 25.95 | 17.51 | 17.12 | 42.87 | 47.45 | 9.48 |

*Table 6.* Performance scaling with number of subjects (out-of-distribution). Evaluation is performed on Subject 8, who is held out during training.

| | | | | | | | | |
|---|---|---|---|---|---|---|---|---|
| number of subjects | BLEU-1 ↑ | BLEU-2 ↑ | BLEU-3 ↑ | BLEU-4 ↑ | METEOR ↑ | ROUGE ↑ | CIDEr ↑ | SPICE ↑ |
| 1 | 42.40 | 19.51 | 8.61 | 4.19 | 9.48 | 30.68 | 5.41 | 1.70 |
| 3 | 43.95 | 22.45 | 10.37 | 5.53 | 10.37 | 32.49 | 5.79 | 2.59 |
| 5 | 45.00 | 22.61 | 10.64 | 5.60 | 10.53 | 32.53 | 6.22 | 2.79 |
| 7 | 47.30 | 25.35 | 13.61 | 8.15 | 11.40 | 34.64 | 6.41 | 3.61 |

ure 6. The results strongly validate our design. The vanilla cross attention (*Pos Enc.+fMRI*) leads to poor performance, while removing fMRI values from the key embeddings (*Pos Enc.*) yields a significant improvement. Replacing positional encoding with region encodings (*Reg. Enc.*) accelerates convergence in the early stages since it is grounded by neuroscientific principles. However, it is eventually outperformed by *Pos Enc.* due to the lack of finer-grained information. Combining the positional encoding and region encodings (*Pos Enc.+Reg Enc.*) achieves the best model design. In addition, replacing positional encoding with an MLP that maps coordinates to embeddings results in poor performance (*(x,y,z)+MLP*), which indicates the amount of high-frequency spatial information in fMRI signals. Additional ablation results on evaluation metrics are provided in Appendix E.

## 4.8. Visualizations and Interpretations

Unlike previous deep learning models (Scotti et al., 2024b; Mai & Zhang, 2023), our model allows interpretations by investigating how *queries* work in the neuroscience-informed attention layer. We inspect the attention weights between queries and voxels in Figure 7.

We found that some queries primarily focus on processing

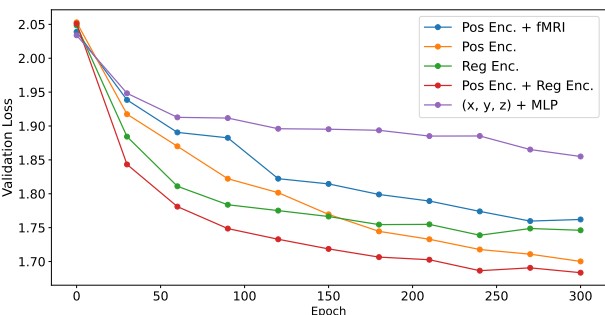

Figure 6. Ablation study of the key embedding design. Pos Enc. stands for positional encoding. Reg Enc. stands for multiple region encodings. *(x, y, z)+MLP* means we employ an MLP to map the coordinates to the embeddings instead of positional encoding.

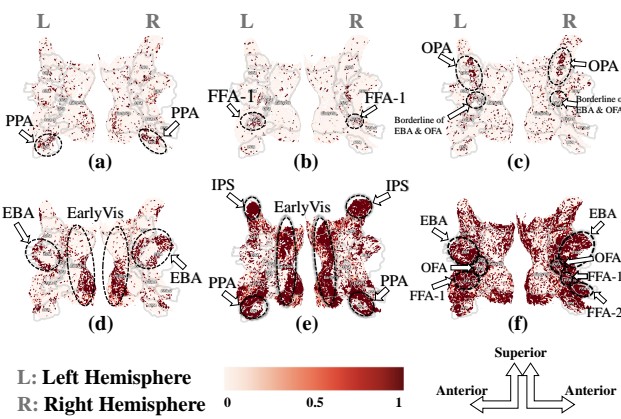

L: Left Hemisphere
R: Right Hemisphere   0      0.5      1

Figure 7. Each subfigure corresponds to a specific query of subject 1. We visualize the attention map between that query token and all voxels. Here we randomly select 6 query tokens, each of which exhibits a distinct spatial focus.

single brain regions, such as Parahippocampal Place Area (PPA) (Figure 7a) and Fusiform Face Area (FFA) (7b). As previous research has shown, PPA is related to conceptual association, semantic processing and environmental memory (Epstein et al., 1999; Köhler et al., 2002; Bar et al., 2008; Epstein & Ward, 2010) and FFA is known for its critical role in expertise recognition, social cognition and identity memory (Schultz et al., 2003; Tsantani et al., 2021; Xu, 2005). Both are important brain regions for the conceptualization of visual information and are responsible for the interaction between real-time stimulus and past memory (Brewer et al., 1998; Ranganath et al., 2004; Golarai et al., 2007).

Moreover, there are some queries that attend to multiple brain regions, revealing the information transmission between low- and high-level brain regions. For instance, interactions between early visual areas and higher-level regions like PPA and IntraParietal Sulcus (IPS) (Figure 7e), revealing a potential pattern for human attention-guided actions

(Tunik et al., 2007; Connolly et al., 2016). Additionally, queries are also found responsible for communications between high-level brain regions (Figure 7c,7f). Together, these findings indicate that the learnable queries may reflect the dynamics of human brain activities in the visual task, from seeing and thinking about the image to pressing the button for the visual recall task in NSD (Allen et al., 2022).

We provide attention map visualizations of subjects other than subject 1 in Appendix I. We also visualize latent embeddings during the model forward pass in Appendix H. Qualitative analysis of model responses could be found in Appendix D.

## 5. Conclusion

In this work, we propose MindLLM, a subject-agnostic and versatile fMRI-to-text decoding model. Our neuroscience-informed attention mechanism in the fMRI encoder ensures subject generalization, while brain instruction tuning enhances versatility. Comprehensive benchmarks demonstrate our model's state-of-the-art performance, and visualizations offer insights into the patterns it leverages. We envision that these advancements will contribute to medical treatments, neuroscience research, and brain-computer interfaces in the future. **Limitations:** This work focuses on static fMRI, without incorporating temporal dynamics. Future research could explore integrating temporal information and investigating its relationship with other modalities such as videos.

## Acknowledgments

We are grateful for funding and computing resource support from Amazon Research, Snap Research, National Science Foundation (NSF) IIS Div Of Information & Intelligent Systems 2403317, Yale AI Engineering Research Seed Grants, Yale Office of the Provost, and Yale Wu Tsai Institute.

## Impact Statement

This paper presents work whose goal is to advance the field of Machine Learning. There are many potential societal consequences of our work, none of which we feel must be specifically highlighted here.

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

# A. Dataset Details

## A.1. Details of each dataset in brain instruction tuning

In this section, we give a brief description of each source of our brain instruction datasets as well as examples from them.

Table 7: Dataset details and examples.

| Dataset | Description | Example | |
|---------|-------------|---------|---|
| Previous Caption | Generating a one-sentence caption of the image that the subject previously saw. |  | A neat bedroom pairs modern chairs with a glass table. |
| COCO Caption (Chen et al., 2015) | Generate a one-sentence caption of the image the subject currently sees. |  | The pedestrian is walking down the side of the highway by the bus. |
| Image Paragraph Captioning (Krause et al., 2017) | Generate a one-paragraph caption of the image the subject currently sees |  | An elephant with a harness, and a seat on his back is seen in a dirt field on a sunny day. The seat on the back of the elephant is brown and is tied with ropes. The sun is shining through the trees onto the ground below. Behind the elephant there is a cinder block wall with grass growing in front of the wall. There are trees behind the wall. |
| COCO QA (Ren et al., 2015) | Answer questions according to the image. |  | [Q]: what eats leaves from a basket in an enclosure? [A]: Giraffe. |
| Visual Genome QA (Krishna et al., 2017) | Answer image-based questions that require richer semantic understanding of the image than COCO-QA. |  | [Q]: Where was the photo taken? [A]: In an office. |
| VQAv2 (Goyal et al., 2017) | Answer image-based questions with better equality and diversity than COCO-QA. |  | [Q]: What are the two white letters? [A]: hu |
| OK-VQA (Marino et al., 2019) | Answer image-based questions that require external knowledge beyond the image itself. |  | [Q]: What part of the body do you wear the rightmost objects on? [A]: Neck. |
| ST-VQA (Biten et al., 2019) | Answer questions of high-level semantic information present in images as the textual cue |  | [Q]: What is written on the front of the shirt? [A]: Marine. |
| TallyQA (Acharya et al., 2019) | Count objects present in images. |  | [Q]: How many bats on the wall? [A]: 2 |
| VQA-E (Li et al., 2018) | Answer questions and generate corresponding explanations for an image-based question. |  | [Q]: Are the people going for a walk in the forest? [A]: Yes. Here is the explanation: A picture of the land, trees, and people passing by as they ride in a vehicle. |
| A-OKVQA (Schwenk et al., 2022) | Answer multiple-choice questions. |  | [Q]: What season is up next? Multiple Choices: A. autumn B. spring C. summer D. winter [A]: B |

| Dataset | Description | Example |
|---|---|---|
| FSVQA (Shin et al., 2016) | Answer the questions in full sentences. |  [Q]: Is this an area that is more populated with pedestrians than cars? [A]: Yes, this is an area that is more populated with pedestrians than cars. |
| VisDial (Murahari et al., 2019) | Generate answers for image-based questions in a multi-turn dialogue. |  [Q1]: Is the photo in color? [A1]: No. [Q2]: Is the man wearing glasses [A2]: Can't tell. [Q3]: How many horses are there? [A3]: 2 |
| LLava Instruction 150K (Liu et al., 2023) | Generate answers for object-level and scene-level answering or reasoning questions for single or multi-round conversations. |  [Q1]: What is the main activity of the person in the fMRI? [A1]: The main activity of the person in the fMRI is brushing her teeth. [Q2]: What should be considered when maintaining oral hygiene in a public setting? [A2]: When maintaining oral hygiene in a public setting (as depicted in the fMRI by a woman brushing her teeth with a green toothbrush while carrying a shoulder bag), factors such as cleanliness, privacy, time management, and personal hygiene supplies should be taken into account... |
| LVIS Instruct4V (Wang et al., 2023a) | Similar to LLava Instruct4V but is finer-grained and of higher quality. |  [Q1]: What is the primary object in the fMRI? [A1]: The primary object in the fMRI is a bird. [Q2]: Can you describe the color and appearance of the bird? [A2]: The bird has a deep brown body with a darker head. Its eyes are distinctively surrounded by a yellowish hue. The bird also possesses a blue-tinted beak. [Q3]: How many birds are in the picture?' [A3]: There is only one bird in the picture... |
| TDIUC (only used in downstream) (Kafle & Kanan, 2017) | Generate answers for image-based questions from 12 different task-directed question types such as object presence and utility/affordance. |  [Q]: What object can be thrown? [A]: Baseball |

## A.2. Dataset statistics

Table 8 summarizes the statistics of each subject in the natural scene dataset (Allen et al., 2022).

*Table 8.* Number of voxels for each subject.

| subject | 1 | 2 | 3 | 4 | 5 | 6 | 7 | 8 |
|---|---|---|---|---|---|---|---|---|
| #(input voxels) | 15724 | 14278 | 15226 | 13153 | 13039 | 17907 | 126682 | 14386 |
| #(samples) | 30000 | 30000 | 24000 | 22500 | 30000 | 24000 | 30000 | 22500 |

## A.3. Instructions

Table 9 lists instructions for each dataset (i.e. $X_{\text{inst}}$). Some instructions are inspired by (Han et al., 2024b). For question-answering tasks, the instructions are appended after the question in a new line.

## B. Implementation details

We choose Vicuna-7b (Zheng et al., 2023) as our backbone LLM. During the brain instruction tuning stage, We use AdamW as the optimizer, with the learning rate of $1 \times 10^{-3}$, weight decay of $0.01$ and $\beta_1 = 0.9$, $\beta_2 = 0.999$. We do not use a learning rate scheduler. We set the batch size to 64. The instruction tuning is conducted on a machine with $8\times$ L40S GPUs for 8 days. And each downstream fine-tuning is conducted on a single L40S GPU with a $1 \times 10^{-4}$ learning rate and 48 batch size. For generations, we have adopted the greedy decoding strategy.

We use 128 as the number of fMRI tokens. We use a 4-layer MLP in the fMRI encoder. We set the number of queries to 1024. The dimension of the query embeddings is 128.

*Table 9.* Instruction template and statistics of the BIT dataset. "−" indicates that the instruction is embedded within the conversation and does not require an additional one. Note that not all conversations are associated with fMRI recordings as only a subset of MSCOCO images were used as stimuli in the study. Consequently, the number of usable conversations in practice will be lower.

| Dataset | #(conversations) | Instruction |
|---|---|---|
| system prompt | / | You are a helpful agent that decodes the brain activity of a person looking at an image. |
| Previous Caption | 149, 875 | Please describe the image the subject saw previously. |
| COCO Caption | 616, 767 | Please describe the fMRI as simply as possible. |
| Image Paragraph Captioning | 9, 598 | Describe the fMRI in one paragraph. |
| COCO QA | 117, 684 | Answer the question with a short phrase. |
| Visual Genome QA | 676, 116 | Answer the question with a short phrase. |
| VQAv2 | 6, 581, 110 | Answer the question with a short phrase. |
| OK-VQA | 140, 550 | Answer the question with a short phrase. |
| ST-VQA | 29, 751 | Answer the question with a short phrase. |
| TallyQA | 238, 056 | Answer the question with a number. |
| VQA-E | 2, 697, 860 | Answer with a short phrase and provide explanation for your answer. |
| A-OKVQA | 18, 201 | Answer with the option's letter from the given choices and provide explanation for your choice. |
| FSVQA | 369, 861 | Answer the question in a full sentence. |
| VisDial | 125, 351 | − |
| Llava Instruct 150K | 157, 712 | − |
| LVIS Instruct4V | 222, 711 | − |

## C. Computational Complexity and Runtime Analysis

In the neuroscience-informed attention, the complexity of the dot product between queries and keys is $O(dNN_q)$. The complexity of the aggregation of values is $O(NN_q)$, which is negligible. The MLP maps the hidden representation of dimension $N_q$ to $L \times d$, therefore its complexity is $dLN_q$. Therefore, the complexity of the fMRI encoder is $O(dNN_q + dLN_q) = O(dN_q(L + N)) = O(dN_qN)$ given that $L \ll N$.

Table 10 summarizes the computational complexity and runtime comparison of different models. Despite its superior performance, our model introduces only a marginal increase in encoder-side computational cost compared to other baselines. Notably, the majority of inference time and complexity arises from the LLM component, which is shared across all models. This highlights that the design choices in the encoder, while crucial for performance, do not significantly affect runtime.

## D. Qualitative Analysis

In this section, we present a qualitative analysis of our model on COCO Captioning, COCO-QA, and OK-VQA, comparing its performance against MindBridge (Wang et al., 2024a) and UniBrain (Wang et al., 2024b). As shown in Figure 8 and 9, our results demonstrate significant improvements in visual understanding across multiple tasks. The model shows strength in the following areas: **Static Object Recognition.** The model demonstrates superior accuracy in identifying stationary objects. In comparison with baseline models (MindBridge and UniBrain), our approach shows improvement in spatial context understanding. For example, when analyzing aircraft imagery (e.g., (a) of Figure 8), our model correctly identifies "airplane sitting on the runway" while baselines incorrectly interpret the scene as "flying in the sky" or "flying over a city", demonstrating better state-space recognition. **Action Recognition.** Our proposed model exhibits enhanced capability in distinguishing between similar actions. In sports scenarios (e.g., (f) of Figure 9), our model correctly identifies "catch ball" while both baselines incorrectly predict "serve", indicating improved action-state discrimination. **Potential of neuroscience application.** The demonstrated improvements in object understanding and action recognition suggest the potential for advancing brain-computer interface technology and neural processing research. The model's enhanced capabilities in distinguishing object states and actions could lead to more effective neural prosthetics and improved assistive technologies for individuals with visual or motor impairments.

*Table 10.* Computational complexity and runtime of each encoder and the LLM. For the first 4 lines, we only time the encoder part, excluding the LLM. The runtime of the LLM forward pass is reported separately in the last row.

| Module | Forward Pass (ms/sample) | Complexity | Notation Explanation | Approximate Values |
|---|---|---|---|---|
| UMBRAE | 0.0109 | $O(dN + dN_q^2)$ | $d$: hidden size
$N$: number of voxels
$N_q$: number of query tokens | $d = 100$
$N = 15000$
$N_q = 200$ |
| MindBridge | 0.0048 | $O(dN)$ | $d$: hidden size
$N$: number of voxels | $d = 1000$
$N = 15000$ |
| UniBrain | 0.0175 | $O(dGK + dN_q^2)$ | $d$: hidden size
$G$: number of groups
$K$: voxels in each group
$N_q$: number of query tokens | $d = 1000$
$G = 500$
$K = 30$
$N_q = 200$ |
| MindLLM | 0.0181 | $O(dN_qN)$ | $d$: hidden size
$N$: number of voxels
$N_q$: number of query tokens | $d = 100$
$N = 15000$
$N_q = 100$ |
| LLM | 0.0621 | - | - | - |

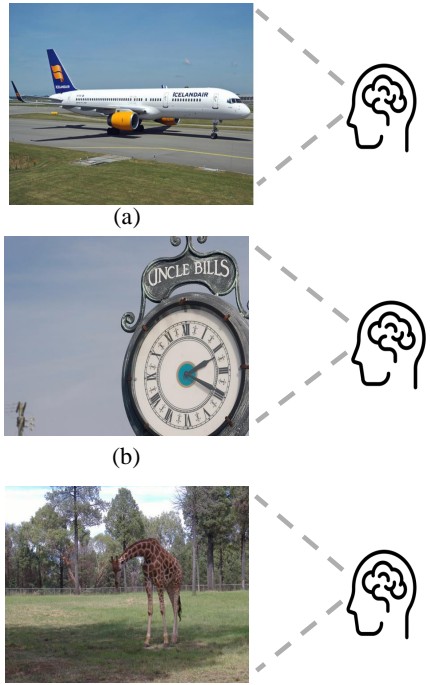

(a)

(b)

(c)

**Ground Truth Caption**: A white jet with blue and orange detailing sits on the runway.
A Icelandair Airliner is preparing for take off.
An airplane sitting at the end of a runway preparing for takeoff.
**Ours**: A large airplane sitting on top of an airport runway.
**MindBridge**: A large passenger jet flying in the sky.
**UniBrain**: A large airplane is flying over a city。

**Ground Truth Caption**: A white jet with blue and orange detailing sits on the runway.
Outdoor clock with metal decoration against a blue sky.
Closeup of a clock with a sign that reads "Uncle Bills".
**Ours**: A clock tower with a clock on it.
**MindBridge**: A sign that says "STOP" on a pole.
**UniBrain**: A red and white airplane is parked on the tarmac.

**Ground Truth Caption**: A giraffe standing on top of a lush green field.
Giraffe standing in middle of fenced in area looking to the ground.
A giraffe standing in the shade of a tree.
**Ours**: A giraffe standing in the middle of a field.
**MindBridge**: A train is traveling down the tracks in the city.
**UniBrain**: A zebra standing in the middle of a street.

*Figure 8.* Qualitative Analysis of COCO Captioning.

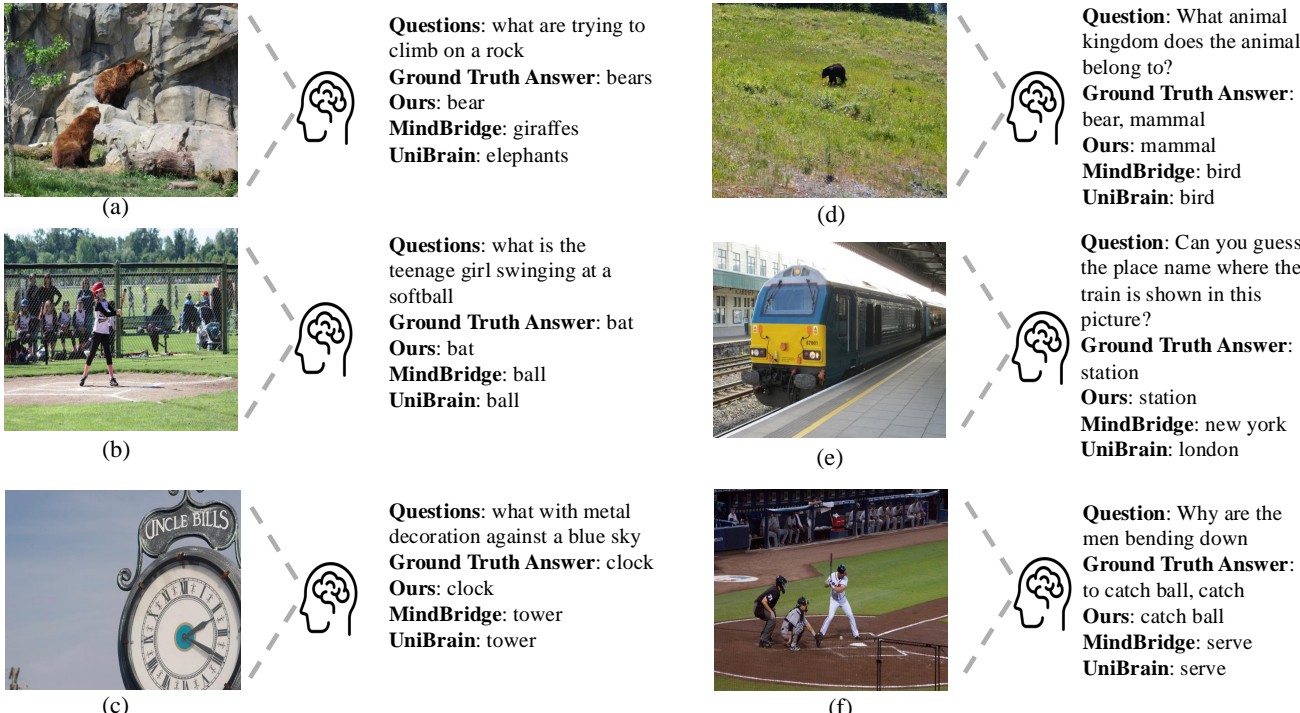

*Figure 9.* Qualitative Analysis of COCO QA (left column) and OK-VQA (right column).

## E. Ablation Studies on Downstream Tasks

Table 11 reports the results of the ablation study on each downstream task. The results validate the design of neuroscience-informed attention.

## F. Architecture Comparison

We make a side-by-side comparison between our model and related baselines in Table 12. Our model is the only one that achieves the four properties listed in the table's head.

## G. Experiments on Other Subjects

We report experiments on subjects 2, 5, 7 in Table 13. We chose these subjects because they are commonly used in previous works. We observe consistent performance gains over the baselines in the majority of scenarios.

## H. Latent Space Visualization

To assess how subject-specific and subject-agnostic information evolves through the model, we visualize the latent embeddings at various stages of the encoder in Figure 10. The original inputs exhibit clear subject-specific clusters. After passing through the neuroscience-informed attention layer, the embeddings remain separable by subject, indicating that subject identity is still preserved. As the data flows through successive layers of the MLP, the subject-specific patterns become increasingly mixed, ultimately forming a subject-agnostic representation. This transformation facilitates generalization on held-out subjects in a zero-shot manner.

## I. Brain Attention Maps of Other Subjects

We visualize the flat brain attention maps (corresponding to Figure 7a) in Figure 11. We observe that the attention maps exhibit highly similar spatial patterns across subjects, indicating that the model captures shared structural-functional

*Table 11.* Ablation study of the key embedding design. Pos Enc. stands for positional encoding. Reg Enc. stands for multiple region encodings. *(x, y, z)+MLP* means we employ an MLP to map the coordinates to the embeddings instead of positional encoding. Note that unlike in Table 2, where we fine-tune the model on each task individually, here we do not fine-tune the model on each individual task and report the performances after brain instruction tuning.

| | | $(x, y, z)$ + MLP | Pos Enc. + fMRI | Pos Enc. | Reg Enc. | Pos Enc. + Reg Enc. |
|---|---|---|---|---|---|---|
| COCO-QA | Accuracy ↑ | 40.57 | 43.14 | **46.29** | 43.90 | 44.76 |
| VG-QA | Accuracy ↑ | 20.29 | 20.39 | **21.12** | 20.83 | **21.12** |
| VQA-v2 | Accuracy ↑ | 45.25 | 46.06 | 47.30 | 47.44 | **47.86** |
| A-OKVQA | Accuracy ↑ | 46.02 | 44.25 | 45.13 | 42.48 | **46.90** |
| ST-VQA | ANLS ↑ | **13.70** | 9.70 | 12.79 | 10.61 | 11.64 |
| Tally QA | Accuracy ↑ | 44.63 | 49.77 | 52.29 | 48.84 | **53.01** |
| | RMSE ↓ | 2.15 | 1.83 | 1.75 | 1.96 | **1.73** |
| COCO-Caption | BLEU-1 ↑ | 49.39 | 54.86 | 57.03 | 52.53 | **58.98** |
| | BLEU-2 ↑ | 26.30 | 33.84 | 37.12 | 33.00 | **39.08** |
| | BLEU-3 ↑ | 14.06 | 20.67 | 23.63 | 20.69 | **25.56** |
| | BLEU-4 ↑ | 8.19 | 13.42 | 15.51 | 13.93 | **17.36** |
| | METEOR ↑ | 12.31 | 14.97 | 16.41 | 14.81 | **17.12** |
| | ROUGE ↑ | 35.45 | 40.17 | 42.14 | 39.62 | **43.34** |
| | CIDEr ↑ | 18.26 | 32.15 | 41.32 | 34.25 | **46.17** |
| | SPICE ↑ | 3.94 | 7.14 | 8.56 | 6.76 | **9.24** |
| Paragraph Caption | BLEU-1 ↑ | 24.13 | 24.57 | 25.90 | 24.74 | **27.17** |
| | BLEU-2 ↑ | 11.88 | 13.03 | 12.41 | 12.61 | **14.50** |
| | BLEU-3 ↑ | 6.12 | 7.16 | 6.10 | 7.20 | **7.99** |
| | BLEU-4 ↑ | 3.55 | 4.24 | 3.22 | 4.35 | **4.67** |
| | METEOR ↑ | 11.19 | 11.78 | 9.05 | 12.12 | **14.43** |
| | CIDEr ↑ | 3.32 | 2.35 | **7.46** | 4.97 | 4.30 |
| VQA-E | Accuracy ↑ | 46.74 | 48.17 | 48.60 | 48.48 | **48.95** |
| | BLEU-1 ↑ | 35.64 | 36.19 | 36.46 | 35.06 | **36.65** |
| | BLEU-2 ↑ | 18.65 | 19.38 | 19.53 | 18.32 | **19.58** |
| | BLEU-3 ↑ | 10.85 | 11.40 | **11.62** | 10.54 | 11.57 |
| | BLEU-4 ↑ | 6.79 | 7.12 | 7.29 | 6.38 | **7.31** |
| | CIDEr ↑ | 83.18 | 88.14 | **89.24** | 85.34 | 88.96 |
| | METEOR ↑ | 14.20 | 14.81 | **15.08** | 14.54 | **15.08** |
| | ROUGE ↑ | 33.81 | 35.15 | **35.17** | 34.28 | 34.76 |
| Average | Perplexity ↓ | 6.39 | 5.82 | 5.47 | 5.73 | **5.38** |

| Method | cross-subject | unified architecture | shared parameters | using all voxels |
|---|---|---|---|---|
| MindEye (Scotti et al., 2024a) | ✗ | ✗ | ✗ | ✓ |
| MindEye2 (Scotti et al., 2024b) | ✓ | ✗ | ✗ | ✓ |
| UMBRAE (Xia et al., 2024) | ✓ | ✗ | ✗ | ✓ |
| MindBridge (Wang et al., 2024a) | ✓ | ✓ | ✗ | ✗ |
| UniBrain (Wang et al., 2024b) | ✓ | ✓ | ✓ | ✗ |
| MindLLM | ✓ | ✓ | ✓ | ✓ |

*Table 12.* Comparison of our model and baselines. Concepts of properties: *cross-subject*: the model performs across subjects; *unified architecture*: the architecture for each subject is the same (the parameters might be different, e.g., MLPs with the same size but different parameters); *shared parameters*: The model uses the same parameters for all subjects; *using all voxels*: the model uses all voxels as inputs, thereby mitigating the risk of information loss from voxel omission. Refer to Section 2.2 and Figure 2 for details of this property.

*Table 13.* Breakdown performances on subjects 2, 5, 7 respectively.

| | | subject 2 | | | | subject 5 | | | | subject 7 | | | |
|---|---|---|---|---|---|---|---|---|---|---|---|---|---|
| | | UMBRAE | UniBrain | MindBridge | MindLLM | UMBRAE | UniBrain | MindBridge | MindLLM | UMBRAE | UniBrain | MindBridge | MindLLM |
| COCO-QA | Accuracy ↑ | 26.85 | 47.14 | 49.04 | **51.71** | 29.86 | 45.52 | 50.57 | **53.14** | 25.78 | 45.05 | 50.19 | **50.76** |
| VG-QA | Accuracy ↑ | 18.65 | 23.53 | 24.97 | **25.96** | 19.38 | 23.65 | 26.11 | **26.89** | 18.44 | 23.02 | 25.16 | **25.92** |
| VQA-v2 | Accuracy ↑ | 49.96 | 48.17 | 51.28 | **53.17** | 51.49 | 48.81 | 51.78 | **53.09** | 50.13 | 47.97 | 50.60 | **52.70** |
| A-OKVQA | Accuracy ↑ | 40.54 | 43.36 | 44.25 | **49.56** | 43.24 | 45.13 | 50.44 | **50.74** | 40.54 | 46.02 | 45.13 | **46.90** |
| ST-VQA | ANLS ↑ | 6.61 | 9.34 | 10.49 | **10.85** | 4.31 | 9.77 | **10.49** | 9.63 | 5.46 | 8.26 | 13.48 | **13.71** |
| OK-VQA | Accuracy ↑ | 20.70 | 32.82 | 33.50 | **35.73** | 22.98 | 30.43 | 34.36 | **35.04** | 17.89 | 34.19 | 34.54 | **34.87** |
| TallyQA | Accuracy ↑ | 42.21 | 55.99 | 0.20 | **57.84** | 42.69 | 54.65 | 58.71 | **59.3** | 40.95 | 53.98 | 56.20 | **58.41** |
| | RMSE ↓ | 4.85 | 1.75 | **1.65** | 1.66 | 4.86 | 1.64 | **1.60** | **1.60** | 3.65 | 1.72 | **1.66** | 1.73 |
| Paragraph Caption | BLEU-1 ↑ | 29.29 | 27.53 | 27.19 | **30.13** | 29.21 | 29.30 | 28.70 | **34.54** | 29.61 | 25.54 | 27.45 | **29.54** |
| | BLEU-2 ↑ | 13.82 | 14.73 | 14.75 | **16.29** | 14.37 | 15.38 | 15.21 | **19.11** | 14.21 | 13.26 | 14.44 | **16.19** |
| | BLEU-3 ↑ | 6.44 | 8.42 | 8.70 | **11.69** | 6.84 | 8.73 | 8.87 | **10.11** | 6.48 | 7.42 | 8.05 | **9.34** |
| | BLEU-4 ↑ | 3.08 | 5.10 | 5.46 | **5.96** | 3.28 | 5.23 | 5.53 | **7.60** | 2.95 | 4.52 | 4.84 | **5.73** |
| | METEOR ↑ | 11.94 | 12.31 | 12.36 | **13.06** | 12.68 | 11.63 | 12.63 | **13.09** | 12.47 | 11.31 | 12.57 | **13.54** |
| | CIDEr ↑ | 5.40 | 7.53 | 0.21 | **11.90** | 7.46 | 6.00 | 8.63 | **12.97** | **8.84** | 4.03 | 5.62 | 8.53 |
| VQA-E | Accuracy ↑ | 47.62 | 50.03 | 52.30 | **54.92** | 49.06 | 50.18 | 54.21 | **55.75** | 48.95 | 49.48 | 53.84 | **54.68** |
| | BLEU-1 ↑ | 30.05 | 36.63 | 37.48 | **38.13** | 30.55 | 37.41 | 38.07 | **38.40** | 29.58 | 36.71 | 38.33 | **38.41** |
| | BLEU-2 ↑ | 14.87 | 19.60 | 20.16 | **20.77** | 14.99 | 19.96 | 20.90 | **21.04** | 14.43 | 19.78 | 20.90 | **20.98** |
| | BLEU-3 ↑ | 8.31 | 11.49 | 11.76 | **12.34** | 8.30 | 11.66 | 12.54 | **12.60** | 7.97 | 11.80 | 12.49 | **12.56** |
| | BLEU-4 ↑ | 4.99 | 7.18 | 7.26 | **7.74** | 4.93 | 7.15 | 7.94 | **8.17** | 4.68 | 7.53 | **7.99** | 7.90 |
| | CIDEr ↑ | 64.60 | 89.50 | 91.93 | **97.68** | 66.47 | 89.91 | 95.30 | **99.48** | 61.34 | 90.49 | 98.23 | **99.36** |
| | METEOR ↑ | 12.37 | 15.12 | 15.56 | **16.02** | 12.51 | 15.37 | 15.88 | **16.28** | 12.09 | 15.02 | 16.07 | **16.16** |
| | ROUGE ↑ | 28.49 | 35.10 | 35.86 | **36.52** | 28.75 | 35.42 | 36.43 | **36.82** | 28.04 | 35.24 | 36.61 | **36.73** |
| FSVQA | VQA Acc. ↑ | 16.37 | 45.62 | 48.62 | **50.19** | 16.40 | 45.66 | 49.45 | **50.76** | 16.37 | 46.11 | 49.19 | **51.13** |
| | FSVQA Acc. ↑ | 0.00 | 40.13 | 43.57 | **45.21** | 0.00 | 45.66 | 44.60 | **45.53** | 0.00 | 40.07 | 44.70 | **46.07** |
| | BLEU-1 ↑ | 18.98 | 86.03 | 87.28 | **87.84** | 19.00 | 85.99 | 87.76 | **87.89** | 18.98 | 85.48 | 87.29 | **88.05** |
| | BLEU-2 ↑ | 9.11 | 81.66 | 83.14 | **83.94** | 11.94 | 81.52 | 83.25 | **83.87** | 10.89 | 81.00 | 83.42 | **84.16** |
| | BLEU-3 ↑ | 1.98 | 77.61 | 79.31 | **80.28** | 3.42 | 77.34 | 79.64 | **80.21** | 2.82 | 76.84 | 79.70 | **80.60** |
| | BLEU-4 ↑ | 0.00 | 73.45 | 75.42 | **76.51** | 1.65 | 73.05 | 75.99 | **76.45** | 0.00 | 72.56 | 76.11 | **77.01** |
| | METEOR ↑ | 5.65 | 48.06 | 49.26 | **49.88** | 5.67 | 48.01 | 49.57 | **50.10** | 5.66 | 47.86 | 49.74 | **50.18** |
| | CIDEr ↑ | 3.19 | 647.62 | 671.26 | **681.94** | 3.26 | 648.21 | 675.99 | **684.12** | 3.37 | 643.44 | 676.50 | **684.72** |

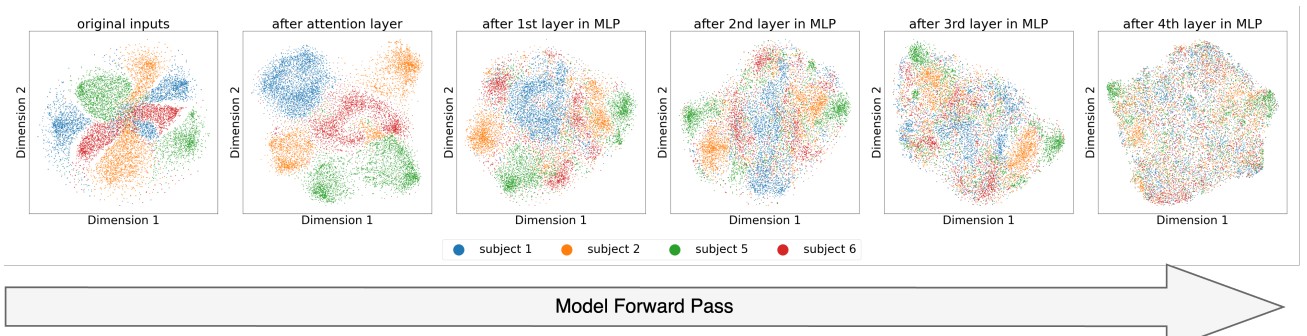

*Figure 10.* Visualization of latent embeddings across model layers using T-SNE.

correspondences across human brains, which is expected due to the overall anatomical and functional similarity among individuals. At the same time, we do observe moderate subject-specific variations in the attention maps. These reflect differences in voxel-level functional signals and individual variability in precise voxel locations. The model is able to accommodate these differences through flexible attention mechanisms.

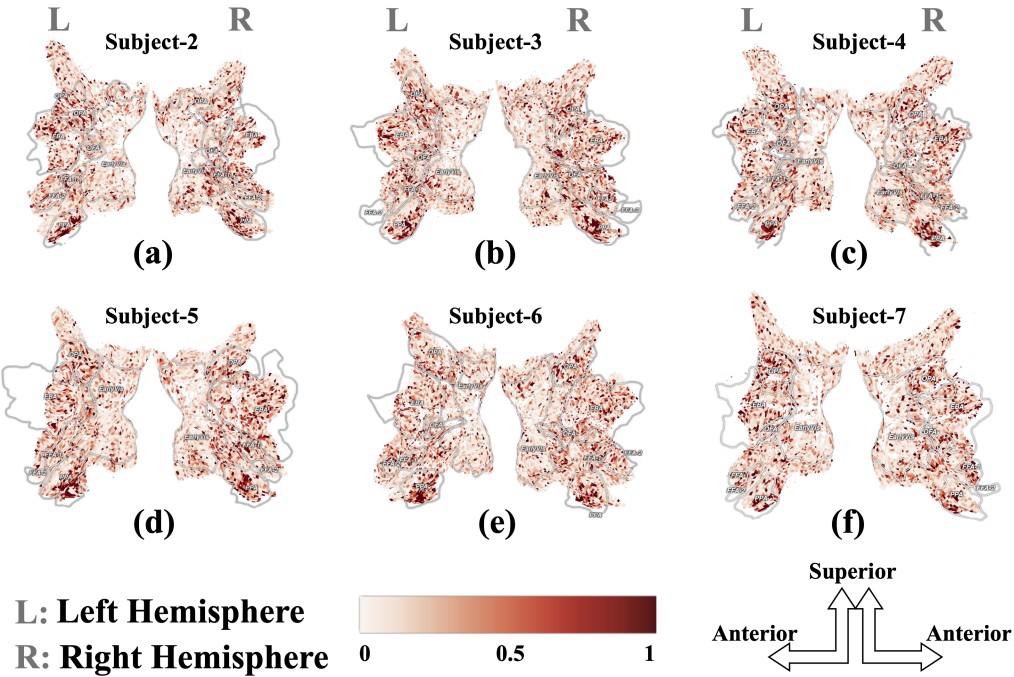

*Figure 11.* Figure 7a's query token on different subjects. It shows similar patterns across subjects, suggesting shared brain organization, while also capturing some subject-specific variances.

