# OpenReview forum: "MindLLM: A Subject-Agnostic and Versatile Model for fMRI-to-text Decoding"
_ICML.cc/2025/Conference — ICML 2025 poster_

### Official Review · Reviewer_D9CN · 2025-03-09

**Overall Recommendation:** 3

**Summary:**

A large body of research in visual image processing has focused on either brain encoding or decoding, aiming to understand how the brain processes natural scenes or reconstructs images from human brain activity. Recent fMRI brain decoding studies have specifically targeted advanced brain-computer interfaces, where incorporating natural language instructions and fMRI encoder latent vectors into a LLM decoder provides deeper insights into brain mechanisms. However, previous studies, where models were trained independently for each subject, have resulted in poor generalization across subjects. To address this, the current study introduces a novel approach called MindLLM. MindLLM develops a subject-agnostic fMRI encoder capable of accommodating subjects with varying input shapes, thus achieving high-performance subject-agnostic decoding. Additionally, the authors introduce brain instruction tuning, which enhances the model’s ability to capture diverse semantic representations from fMRI signals and generalize to new tasks. Experimental results on the popular NSD (Natural Scenes Dataset) and comprehensive fMRI-to-text benchmarks demonstrate that MindLLM outperforms baselines in downstream tasks, unseen subject generalization, and novel task adaptation.

**Claims And Evidence:**

The submission makes several claims about the effectiveness and innovation of the MindLLM approach, particularly in its subject-agnostic encoding and the ability to generalize across subjects. However, some of these claims are not fully supported by clear and convincing evidence:
* The authors claim that the encoder is subject-agnostic is made, but the evidence provided is not sufficiently detailed to demonstrate how the shared and unique information across subjects is learned. There is a need for more explicit evidence on how the encoder generalizes across subjects, especially given that different subjects have varying numbers of voxels. The absence of a thorough analysis of this aspect makes the claim less convincing.
* While the paper presents results on multiple benchmarks, there is no clear distinction between subject-generalizable benchmarks and those that show subject-specific variations. It would be beneficial to see a more detailed analysis of the performance on individual subjects, and whether the model performs similarly across all subjects on certain tasks or whether it varies significantly. This would help substantiate the claim that MindLLM generalizes effectively.
* The authors claim that brain instruction-tuning enhances the model’s performance, but the details on how this is performed (e.g., whether the loss is backpropagated, and how it affects the LLM decoder’s weights) are not provided. The lack of a comparison between results before and after brain instruction-tuning weakens the claim, as it is not clear how much improvement the tuning provides or what specific changes occur in the model's performance due to this process.
* The authors claim that the flatbrain maps illustrate important insights from the model is undermined by the difficulty in interpreting the figure due to the colormap choice. Without a clearer visualization and better explanation of what each flatmap represents, particularly in relation to specific subjects and query tokens, the evidence provided in Figure 6 does not strongly support the claims about the model’s effectiveness.

**Essential References Not Discussed:**

Yes, all the relevant works are clearly discussed in the current version, and the related work section is adequate.

**Experimental Designs Or Analyses:**

The experimental design of the study is generally sound, but there are several areas that need further clarity and validation. These issues are highlighted in the weaknesses section.

**Methods And Evaluation Criteria:**

The proposed methods, including the subject-agnostic encoder and brain instruction-tuning, are suitable for the problem of brain decoding and the application of multimodal language models. However, additional clarifications on how the methods are implemented and evaluated (particularly in terms of how they handle varying voxel counts and the impact of brain instruction-tuning) would enhance the robustness of the approach. The evaluation criteria, including the use of fMRI-to-text benchmarks, are appropriate, but further breakdowns of subject-specific performance would provide a more comprehensive understanding of the model's generalizability.

**Other Comments Or Suggestions:**

* Clarity of Figures: Some figures, especially Figure 6 (flatbrain maps), could benefit from clearer labeling, improved colormap choices, and better explanations to enhance readability and interpretation.
* The authors could consider including subject-specific and shared variance brain maps to further illustrate the model's ability to generalize across subjects and capture common brain activity patterns.

**Other Strengths And Weaknesses:**

Strengths:
1. The concept of learning a query weight matrix to transform each subject's fMRI data into tokens is intriguing, as this approach can be generalized for new subjects with varying input shapes.

2. The method of handling position encoding for key vectors, particularly the use of Fourier transformers for voxel coordinates, is well-handled.

3. The proposed MindLLM is interesting, as it leverages fMRI-to-text benchmarks and utilizes fMRI encoding vectors passed to an LLM decoder, making MindLLM an fMRI instruction-tuned multimodal large language model.

Weaknesses:

1. There are several significant weaknesses in this work, particularly regarding the methodology and experimental results:
   * The majority of the results presented in this study focus on fMRI-text decoding benchmarks. However, the underlying methodology—the subject-agnostic fMRI encoder—has not been sufficiently explained. Specifically, how was this approach implemented, and was the latent space verified for each subject? Given that fMRI recordings have low signal-to-noise ratio (SNR), it is crucial to understand how the common latent space functions. Furthermore, whether the transition from the latent space to the subject space is accurately reconstructing the voxels and regions is a key issue. Without proper validation of this, it is difficult to trust the results of the fMRI-text decoding.
   * Additionally, the authors should clarify their subject-agnostic fMRI encoder approach in more detail. With the current explanation, it is unclear how the same query weight matrix is learned across subjects. For example, if one subject has 12,000+ voxels and another has 15,000 voxels, how is the same weight matrix used to project data into a common space? If I misunderstood, the authors should provide a clearer explanation of this methodology.

2. The concept of Mind-LLM with a subject-agnostic encoder is not entirely novel. It seems quite similar to the approach presented in the MindEye-2 paper, where the authors also learn a subject-agnostic encoder and use it to generalize on held-out subjects. Therefore, the authors of the current study could provide a clearer explanation of how the MindLLM approach differs from MindEye-2, especially considering that both papers use the same NSD dataset.

3. There are no details provided on how the authors perform brain instruction-tuning. Specifically, it is unclear whether the loss is backpropagated during brain instruction-tuning. If so, how are the weight parameters in the LLM decoder affected or updated? Additionally, there is no comparison of the results before and after brain instruction-tuning to demonstrate its impact.

4. Since the authors have learned a subject-agnostic encoder, it would be helpful to understand what specific shared information is learned across subjects and what unique information is learned for each subject. Additionally, as the authors perform various benchmarks, it would be insightful to know if any benchmarks exhibit similar performance across subjects, while others show subject-specific variations in results.

5. What are the key conclusions from MindLLM? The flatbrain maps in Figure 6 are difficult to interpret due to the choice of colormap. Additionally, there are six flatmaps, and it is unclear whether each subfigure corresponds to a specific subject. The authors mention that each subfigure is related to a query token, but this makes Figure 6 difficult to read and interpret. As a result, drawing meaningful conclusions from this figure is challenging.

6. The paper lacks a discussion of the findings and implications of the current work.

**Questions For Authors:**

* The questions raised in the weaknesses section regarding the subject-agnostic encoder, brain instruction-tuning, and performance across subjects would benefit from further clarification.
Please refer to the details discussed in that section for more context.

**Relation To Broader Scientific Literature:**

The key contributions of the MindLLM approach build upon several important areas of scientific literature, including brain encoding and decoding, brain-computer interfaces, multimodal llms, and cross-subject generalization.

**Theoretical Claims:**

Since the paper does not present formal mathematical proofs to substantiate its theoretical claims, the correctness of any such proofs does not apply.

---

> ### Author Rebuttal · Authors · 2025-04-01
>
> **C1** It is unclear how the fMRI encoder handles different numbers of voxels.
>
> **R1** Each voxel is treated as a token in our model, and the attention layer learns to map sequences of varying lengths into a fixed-dimensional representation. This is similar to using multiple [CLS] tokens in a BERT model to capture the semantics of sentences of various lengths. This works because, in attention, the output dimension is solely determined by the number of queries. We also provide a [mathematical clarification](https://anonymous.4open.science/api/repo/MindLLM-FAA4/file/rebuttal/math_clarification.png)
>
> **C2** The idea is not entirely novel—see MindEye2.
>
> **R2** We respectfully disagree with the statement and would like to emphasize the **fundamental differences** between our approach and MindEye2:
>
> 1. MindEye2 requires a separate projector layer trained on each subject to map their features into a shared representation space.
> 2. The architecture of MindEye2 cannot handle subjects with varying numbers of voxels.
>
> Therefore, MindEye2 cannot generalize to unseen subjects without further tuning on data from the unseen subject. In contrast, our model does not need subject-specific parameters or architectures and can generalize to held-out subjects in a zero-shot manner.
>
> **C3** The brain instruction tuning lacks details of backpropagation.
>
> **R3** The brain instruction tuning trains the model through the loss function in lines 254-255. The gradients are backpropagated from the predicted tokens to the LLM, and all the way back to the fMRI encoder to update its parameter.
>
> **C4** No comparison to show the results w/ and w/o the brain instruction tuning.
>
> **R4** We would like to politely point out that we did have a comprehensive experiment to show the effects of brain instruction tuning (BIT). As shown in Table 2’s caption, models with $\circ$ are the versions w/o BIT. We can see models w/ BIT significantly outperform their corresponding models w/o BIT. On average, BIT brings 28.0% improvement, as stated in line 340, right column.
>
> **C5** Figure 6 is difficult to interpret.
>
> **R5** We improved [Figure 6](https://anonymous.4open.science/api/repo/MindLLM-FAA4/file/rebuttal/figure6.png)'s clarity by selecting a more perceptually friendly colormap. We also improved the caption to better clarify that each subfigure corresponds to a query token on subject 1. The findings are discussed in section 4.7.
>
> **C6** Brain maps across subjects should be included.
>
> **R6**
> We thank the reviewer for the suggestion. We now present the query token from Figure 6(a) applied to additional subjects (Subjects 2–7) below.
>
> [Figure](https://anonymous.4open.science/api/repo/MindLLM-FAA4/file/rebuttal/brainmaps.png)
>
> We observe that the attention maps exhibit highly similar spatial patterns across subjects. This consistency supports the generalizability across subjects to capture common brain patterns.
>
> **C7** The shared vs. subject-specific information in the latent space should be verified.
>
> **R7** We do not encourage the encoding of subject-specific information in the method, as our model is designed to be *subject-agnostic.* This enables scalable deployment across diverse populations and eliminates the need for costly, subject-specific calibration.
>
> To assess how subject-specific and subject-agnostic information evolve through the model, we visualize the latent embeddings at various stages of the encoder below.
>
> [Figure](https://anonymous.4open.science/api/repo/MindLLM-FAA4/file/rebuttal/latent.png)
>
> **C8** Performances on individual subjects should be included.
>
> **R8**
> We appreciate the suggestion. The results in Table 2 are based on subject 1. We have now extended our experiments to include subjects 2 and 5 below.
>
> [Table](https://anonymous.4open.science/api/repo/MindLLM-FAA4/file/rebuttal/breakdown.png)
>
> Across subjects, our model outperforms the baselines in most cases, demonstrating robustness to inter-subject variability. We note that performance varies slightly more on ST-VQA and TallyQA. Models trained on subject 5 outperform those trained on subjects 1 and 2, suggesting that subject 5 provides higher-quality data.
>
> We will include results for all subjects in the revised version.
>
> **C9** What are the key conclusions from MindLLM? The paper lacks a discussion of the findings and implications of the current work.
>
> **R9** While interpretation (Figure 6) supports our model’s motivation, our primary focus is **improving decoding performance**.
>
> Key findings:
>
> - **Brain instruction tuning** significantly improves the performance.
> - **Neuroscience-informed attention** significantly outperforms vanilla cross-attention (Section 4.6), offering architectural insights for fMRI decoding.
>
> Impact:
>
> Our method enables subject-agnostic decoding and easy task adaptation, which unlocks out-of-the-box applications like prosthetic control without subject-specific finetuning. We will update detailed discussions in the revised version.

---

> > ### Comment · Reviewer_D9CN · 2025-04-04
> >
> > I thank the authors for addressing several of my questions, particularly clarifying how MindLLM differs from the MindEye-2 paper. The explanation of how each voxel is mapped into an embedding vector by treating it as a token is also very clear. I appreciate the adjustments made to the figures, which have improved readability.
> >
> > However, further clarification regarding the encoder architectures and training paradigms—ideally through a brief tabular or side-by-side architectural comparison—would help highlight the distinctions between MindEye-2 and MindLLM more clearly. Additionally, an analysis of shared versus subject-specific variance would strengthen the paper’s claims.
> >
> > Therefore, I am raising my score in recognition of the improved clarity on several important points.

---

> > > ### Author Response · Authors · 2025-04-06
> > >
> > > We thank the reviewer for recognizing the contributions of our work and for providing additional thoughtful suggestions.
> > >
> > > **CC1**
> > >
> > > The distinction between the proposed model and MindEye2 should be highlighted through a brief tabular or side-by-side architectural comparison.
> > >
> > > **RR1**
> > >
> > > We have 1) updated a side-by-side comparison between our model and MindEye2; 2) a table summarizes the distinctions between our model and all important baselines below
> > > [Figure & Table](https://anonymous.4open.science/api/repo/MindLLM-FAA4/file/rebuttal/comparison.png)
> > >
> > > **CC2**
> > >
> > > There should be additional analysis of shared versus subject-specific variance.
> > >
> > > **RR2**
> > >
> > > We further updated the [flatbrain maps](https://anonymous.4open.science/api/repo/MindLLM-FAA4/file/rebuttal/brainmaps.png) to include quantitative measurements. Specifically, given that voxels differ for different subjects, we propose [spatially weighted cosine similarity (SWCS)](https://anonymous.4open.science/api/repo/MindLLM-FAA4/file/rebuttal/SWCS.png), a metric between $[-1, 1]$ to assess similarities between the attention maps of different subjects. As a baseline for comparison, we also generate uniform random attention maps for each subject and compute their pairwise similarities.
> > >
> > >  We further analyze the results below based on the updated flatbrain maps.
> > >
> > > 1. We observe that the attention maps exhibit highly similar spatial patterns across subjects, indicating that the model captures shared structural-functional correspondences across human brains, which is expected due to the overall anatomical and functional similarity among individuals.
> > > 2. At the same time, we do observe moderate subject-specific variations in the attention maps. These reflect differences in voxel-level functional signals and individual variability in precise voxel locations. The model is able to accommodate these differences through flexible attention mechanisms.
> > > 3. Based on 1 & 2, we argue that the neuroscience-informed attention layer in our model learns both a global understanding of brain organizations and performs finer-grained alignment across subjects when, for example, voxel $i$ in subject 2 may correspond functionally to voxel $j$ in subject 3. However, the actual fMRI signals at corresponding locations may still vary in distribution across subjects.
> > > 4. To address this, our model includes subsequent MLP blocks that further transform the attention-aggregated voxel representations to a shared space, eliminating subject-specific variance, as illustrated in the  [Figure](https://anonymous.4open.science/api/repo/MindLLM-FAA4/file/rebuttal/latent.png).
> > >
> > > We also plan to conduct more comprehensive analyses regarding this topic in the revised version and in future work.

---

### Official Review · Reviewer_SE3R · 2025-03-13

**Overall Recommendation:** 3

**Summary:**

This works a versatile and subject-agnostic model for fMRI-to-text decoding. It demonstrated a brain-instruction tuning approach, inspired from visual instruction tuning framework. This model has specifically designed with application across subjects with varied number of recording voxels, which is a common challenging question in the BCI applications. Compared to existing model using an averaging pooling or sampling strategy, this model proposes an neuroscience-informed attention structure that allows information extraction effectively. In the encoder, the query embeddings are learnable, and key embedding integrate positional information. Meanwhile, it  utilized the NSD datasets (MSCOCO datasets) to build the link between visual images and text inputs, which allows diverse and versatile downstream tasks like perception, memory, language, and reasoning.

**Claims And Evidence:**

This framework is subjective-agnostic, as the designed attention modules allow flexible information extraction regardless of the mismatch of recorded voxels, the claim is clearly supported. As it is demonstrated in a comprehensive benchmarks, compared to existing methods, MindLLM achieves the state-of-the-art accuracy in different downstream tasks. Meanwhile, it has also been evaluated on a subject held out settings, and achieving the best accuracy.

**Essential References Not Discussed:**

Although this work only focused on text decoding, while it lacks of comparisons or references on brain-machine alignment or image/video decoding from related literatures, for example [1][2][3].

[1] Interpreting and improving natural-language processing (in machines) with natural language-processing (in the brain).

[2] Cinematic Mindscapes: High-quality Video Reconstruction from Brain Activity.

[3] Seeing beyond the brain: Conditional diffusion model with sparse masked modeling for vision decoding.

**Experimental Designs Or Analyses:**

This experiment designs are solid, as it has been demonstrated on largest-scale brain decoding dataset recently released, although including other datasets such as BOLD5000 (cross-dataset generalization) could increase more soundness of the evaluation. Multiple established baselines are compared against. An ablation study of critical components in the models has also been implemented in this study.

**Methods And Evaluation Criteria:**

NSD dataset is a well-established benchmark dataset in brain decoding, and MSCOCO dataset is a popular dataset in computer vision, and has been extended into other diverse task settings, including captioning, QA, etc. The evaluations and metrics including BLEU, METEOR are properly utilized. Multiple baselines in this field including MindBridge, UniBrain, BrainChat are compared against.

**Other Comments Or Suggestions:**

N/A

**Other Strengths And Weaknesses:**

This work demonstrate an impactful framework for brain instruction tuning, and allows subject-agnostic decoding, and achieves the state-of-the-art accuracy, while it does not outperforms existing baselines significantly. Meanwhile the framework itself is not novel, as it directly mimic the visual instruction tuning. The framework is lack of capability of integrating the image information, which potentially could be a future extension.

**Questions For Authors:**

1. Computational cost for model's training and inference, compared to other baselines.
2. Does scaling laws of data observed in the results, more subjects trained will increase the accuracy?
3. How is subject-specific information encoded in this framework?

**Relation To Broader Scientific Literature:**

Brain decoding is a challenging field, recent studies start to integrate brain decoding with existing large language models framework to enhance its capability for text decoding. Subject-agnostic decoding is one of the major challenges in this field, and this work is well-motivated to address such issue. This work covers sufficient literature review, is comprehensively grounded with comparisons with multiple existing baseline model, and the results demonstrate certain level of improvement.

**Theoretical Claims:**

This work does not include any theoretical proofs or claims.

---

> ### Author Rebuttal · Authors · 2025-04-01
>
> **C1**
>
> The manuscript lacks comparisons or references on brain-machine alignment or image/video decoding [1][2][3].
>
> **R1**
>
> We thank the reviewer for pointing out these relevant references. [1] [2] deal with fMRI time series, while our method focuses on static fMRI (i.e., fMRI signals at a moment). Therefore, they are not comparable to our work. We will create a new section for multimodal brain decoding in related work and discuss these models in the revised version.
> For [3], we have included it in the baselines. The results are shown below and will be updated in the revised version (The MindVis column is the new baseline).
>
> [Table](https://anonymous.4open.science/api/repo/MindLLM-FAA4/file/rebuttal/table2.png)
>
> **C2**
>
> The framework is not very novel, as it directly mimics the visual instruction tuning.
>
> **R2**
>
> The training paradigm of brain instruction tuning is similar to visual instruction tuning. However, the construction of the brain instruction tuning datasets is a nontrivial contribution. As discussed in section 3.3 (lines 177-184, right column),  we select datasets to capture diverse aspects of semantic information embedded in fMRI signals, which are considered among the most fundamental and essential properties of human brains.
>
> **Q1**
> The computational cost for model's training and inference, compared to other baselines.
>
> **A1**
>
> We thank the reviewer for this valuable question. We summarize the computational complexity and runtime analysis below.
>
> [Table](https://anonymous.4open.science/api/repo/MindLLM-FAA4/file/rebuttal/cost.png)
>
> Experiments here were conducted on run on an A40. All timings are measured during inference with `batch_size=1`. When comparing with other encoders, we only measure the encoder part (LLM is excluded). We separately time the LLM at the last row.
>
> Despite its superior performance, our model introduces only a marginal increase in encoder-side computational cost compared to other baselines. Notably, the majority of inference time and complexity arises from the LLM component, which is shared across all models. This highlights that the design choices in the encoder—while crucial for performance—do not significantly affect runtime.
>
> **Q2**
>
> Does scaling laws of data observed in the results, more subjects trained will increase the accuracy?
>
> **A2**
>
> We appreciate the reviewer’s valuable question and suggestion. We conducted experiments to evaluate how model performance scales with the number of subjects and report the performances on the COCO caption task. We examined both in-distribution (seen subjects) and out-of-distribution (held-out subjects) settings.
>
> [Table](https://anonymous.4open.science/api/repo/MindLLM-FAA4/file/rebuttal/scaling.png)
>
> Our show significant performance improvements as the number of training subjects increases, demonstrating that the model benefits from exposure to more subjects. We will include more results on other subjects and datasets in the revised version.
>
> **Q3**
>
> How is subject-specific information encoded in this framework?
>
> **A3**
>
> We do not encourage the encoding of subject-specific information in the method, as our model is designed to be *subject-agnostic*—capable of generalizing to individuals not seen during training without requiring additional fine-tuning. This "out-of-the-box" generalization has significant practical benefits: it supports scalable deployment across diverse populations and eliminates the need for costly, subject-specific personalization or calibration.
>
> To assess how subject-specific and subject-agnostic information evolve through the model, we visualize the latent embeddings at various stages of the encoder using T-SNE below.
>
> [Figure](https://anonymous.4open.science/api/repo/MindLLM-FAA4/file/rebuttal/latent.png)
>
> This figure illustrates the transition from subject-specific to subject-agnostic representations. Initially, the embeddings exhibit distinct subject-wise clusters, indicating strong subject-specific information. After passing through the neuroscience-informed attention layer, these clusters are still visible. However, as the embeddings propagate through successive MLP layers, the subject-specific patterns gradually dissolve. By the final layer, the latent space forms a well-mixed representation where subject identity is no longer distinguishable.
>
> This transformation is crucial for enabling generalization to unseen subjects in a zero-shot manner.

---

### Official Review · Reviewer_1cT4 · 2025-03-14

**Overall Recommendation:** 4

**Summary:**

The paper proposes a subject-agnostic encoding from fMRI recordings into an LLM space to enable text decoding from brain data. The paper claims this approach generalizes across subjects with different numbers of voxel measurements, and that it outperforms existing baselines.

**Claims And Evidence:**

As written in Table 1, the proposed MindLLM is the only model that is subject-agnostic. But only a few lines above in the text, it is written that UniBrain (Mai & Zhang, 2023) is capable of doing so: "models for fMRI decoding can not handle varying input shapes and are not subject-agnostic, with only a few exceptions (Mai & Zhang, 2023)". Which is correct?

Model performance indeed seems superior to baseline methods, with consistent gains across datasets (Tables 1 and 2). The generalization across subjects is what is most convincing to me (Table 3). The model also seems to adapt well to new tasks (Table 4).

Overall, this seems like a good contribution.

**Essential References Not Discussed:**

N/A.

**Experimental Designs Or Analyses:**

Using standard benchmarks as far as I can tell. I cannot judge if the choice of baselines is comprehensive.

**Methods And Evaluation Criteria:**

Evaluation methods are fair as I can tell, but I am not an expert in brain-to-text decoding.

**Other Comments Or Suggestions:**

Line 32 "results" should be upper-case at beginning of sentence. Something is also off about the grammaticality of the sentence.

**Other Strengths And Weaknesses:**

-

**Questions For Authors:**

How well does your model scale with the number of subjects? I.e. how much better does it perform with pre-training on e.g. 3 subjects compared to 1 etc?

edit after rebuttal: authors have run this and results look very promising. Increased score from 3 to 4.

**Relation To Broader Scientific Literature:**

Decoding text signals from brain data is a broadly studied phenomena, and the use of LLMs for brain data has gained a lot of popularity in recent years. This work adds to that body of literature by proposing a novel subject-agnostic encoder to adapt individual brain recordings into a common LLM space.

**Theoretical Claims:**

N/A.

---

> ### Author Rebuttal · Authors · 2025-04-01
>
> **C1**
>
> The text claims UniBrain is subject-agnostic, but Table 1 lists MindLLM as the only subject-agnostic model—this inconsistency needs clarification.
>
> **R1**
>
> We apologize for the mistake and thank the reviewer for bringing this to our attention.
>
> > with only a few exceptions (Mai & Zhang, 2023).
> >
>
> Should actually be
>
> > with only a few exceptions (Wang et al., 2024b)
> >
>
> The models in the two references share the same name, which led to the citation error. We will correct it in the revised version.
>
> **C2**
>
> Line 32 "results" should be upper-case at the beginning of the sentence. Something is also off about the grammaticality of the sentence.
>
> **R2**
>
> We thank the reviewer for pointing this out. We have corrected the capitalization and revised the sentence for clarity and grammatical correctness. The updated version reads:
>
> > The results demonstrate that our model outperforms the baselines in downstream tasks, generalization to unseen subjects, and adaptation to novel tasks.
> >
>
> **Q1**
>
> How well does your model scale with the number of subjects? I.e. how much better does it perform with pre-training on e.g. 3 subjects compared to 1 etc?
>
> **A1**
>
> We appreciate the reviewer’s valuable question and suggestion. We conducted experiments to evaluate how model performance scales with the number of subjects and report the performances on the COCO caption task. We examined both in-distribution (seen subjects) and out-of-distribution (held-out subjects) settings.
>
> [Table](https://anonymous.4open.science/api/repo/MindLLM-FAA4/file/rebuttal/scaling.png)
>
> Our results show significant performance improvements as the number of training subjects increases, demonstrating that the model benefits from exposure to more subjects during pre-training. We will include more results on other subjects and datasets in the revised manuscript.

---

> > ### Comment · Reviewer_1cT4 · 2025-04-02
> >
> > Thank you for your responses.
> >
> > > As written in Table 1, the proposed MindLLM is the only model that is subject-agnostic. But only a few lines above in the text, it is written that UniBrain (Mai & Zhang, 2023) is capable of doing so: "models for fMRI decoding can not handle varying input shapes and are not subject-agnostic, with only a few exceptions (Mai & Zhang, 2023)".
> >
> > > "with only a few exceptions (Mai & Zhang, 2023)." Should actually be "with only a few exceptions (Wang et al., 2024b)"
> >
> > I'm still confused, I was more pointing to the inconsistency between Table 1 and text, less so the citation. It seems that MindLLM is then **not** the only model that is subject-agnostic. So is Table 1 incorrect?
> >
> > > Our results show significant performance improvements as the number of training subjects increases, demonstrating that the model benefits from exposure to more subjects during pre-training. We will include more results on other subjects and datasets in the revised manuscript.
> >
> > I honestly find this the most interesting result of the paper. I'll increase my score, I think this is worth sharing with the community.

---

> > > ### Author Response · Authors · 2025-04-03
> > >
> > > Thanks for the follow-up. We now better understand the source of confusion and appreciate the opportunity to clarify further.
> > >
> > > UniBrain (Mai & Zhang, 2023) is not subject-agnostic, and has been included in Table 1.
> > >
> > > UniBrain (Wang et al., 2024b) is subject-agnostic, and has been included in **Table 2, 3 & 4**, but not in Table 1. *It is worth noting that although it is subject-agnostic, it exhibits limitations as discussed in lines 98-105 (right column) and illustrated in Figure 2.*
> > >
> > > The reason UniBrain (Wang et al., 2024b) was not originally included in Table 1 is that Table 1 is a widely-used public captioning benchmark, which has been adopted by many prior works.  We only included baselines that reported results on this benchmark in their **original publications**. Since UniBrain (Wang et al., 2024b) was not designed for captioning and did not report results on this benchmark, it was excluded.
> > >
> > > However, to provide a more complete comparison, we have now adapted UniBrain (Wang et al., 2024b) to this benchmark (as we did in Tables 2–4) and present the updated Table 1 below:
> > >
> > > [Table 1](https://anonymous.4open.science/api/repo/MindLLM-FAA4/file/rebuttal/caption.png)
> > >
> > > In conclusion, both the text (with corrected citations) and Table 1 are correct. It’s just that (Wang et al., 2024b) (i.e., the *exception*) was not in Table 1.

---

### Official Review · Reviewer_Gvse · 2025-03-15

**Overall Recommendation:** 2

**Summary:**

This paper proposes MindLLM for subject-agnostic and versatile fMRI-to-text decoding. MindLLM consists of an fMRI encoder and an off-the-shelf LLM. The paper evaluates MindLLM on several fMRI-to-text benchmarks.

**Claims And Evidence:**

The paper claims "a voxel’s position alone can theoretically serve as effective keys for attention weight computation", but no evidence is provided. It seems more like a guess rather than a fact. It is important to verify the rationale of this operation.

**Essential References Not Discussed:**

None.

**Experimental Designs Or Analyses:**

- It is not clear how much of the improvement is due to the proposed model versus the use of superior LLMs. It is unfair to compare their performances with other state-of-the-art baselines that do not use the same LLM in Table 1. The authors should report the performance of their runs using the same backbones as the compared methods for a fair comparison.
- Table 2/3 have similar issues to Table 1.
- In the captioning task in Table 1, the METEOR, CIDEr and SPICE metrics are often considered more important. However, the results for these metrics of the proposed method are only slightly superior to or even inferior to the baselines, which raises doubts about the effectiveness of the proposed method.
- The ablation study in section 4.6 is not sufficiently convincing. There is no explanation of the task being experimented on, and the authors should provide the final results, not just the loss.

**Methods And Evaluation Criteria:**

See the above.

**Other Comments Or Suggestions:**

None.

**Other Strengths And Weaknesses:**

See the above problems.

**Questions For Authors:**

None.

**Relation To Broader Scientific Literature:**

No.

**Theoretical Claims:**

Not applicable.

---

> ### Author Rebuttal · Authors · 2025-04-01
>
> **C1**
>
> The paper claims "a voxel’s position alone can theoretically serve as effective keys for attention weight computation", but no evidence is provided.
>
> **R1**
>
> We would like to point out politely that it is not a guess—it is strongly supported by the ablation study (note the blue line) in section 4.6. As pointed out in lines 403-406, left column,
>
> > The vanilla cross attention (Pos Enc. + fMRI) leads to poor performance, while removing fMRI values from the key embeddings (Pos Enc.) yields a significant improvement.
> >
>
> which validates the hypothesis of using a voxel’s position alone.
>
> **C2**
>
> For Tables 1, 2 & 3, It is not clear how much of the improvement is due to the proposed model versus the use of superior LLMs. It is unfair to compare their performances with other state-of-the-art baselines that do not use the same LLM in Table 1.
>
> **R2**
>
> We argue that the comparison is fair across Table 1-3, as pointed out in lines 140-142
>
> > In practice, we use Vicuna-7b (Zheng et al., 2023) as our LLM to maintain consistency with our baseline (Xia et al., 2024).
> >
>
> Specifically,
>
> - **Table 1**: The strongest baseline, UMBRAE, uses Shikra-7b, a vision-language model improved based on Vicuna-7b—consistent with our choice of LLM.
> - **Table 2**: Besides UMBRAE, for other competitive baselines (MindBridge, UniBrain), we use the same LLM backbone as ours.
> - **Table 3**: All baselines share the same LLM backbone as our model.
>
> Furthermore, as discussed in Section 4.2, we observe substantial performance gains attributable to the **brain instruction tuning** and the **encoder design**. This supports our claim that the improvements stem from our proposed approach rather than solely from the choice of LLM.
>
> **C3**
>
> The results for METEOR, CIDEr and SPICE of the proposed method are only slightly superior to or even inferior to the baselines.
>
> **R3**
>
> While our model does not significantly outperform the baselines in terms of METEOR, CIDEr, and SPICE, it offers several key advantages that the baselines lack:
>
> - As shown in Table 1, our model is the only one that is subject-agnostic. In other words, it can generalize to unseen subjects in a zero-shot manner. This is particularly valuable in BCI applications, where users often expect the device to work out of the box without requiring subject-specific training data.
> - As noted in lines 82–83 (right column), our model can handle tasks beyond simple image-stimulus associations—such as answering memory-related questions—while the strongest baseline, UMBRAE, cannot.
>
> We would also like to emphasize that our method demonstrates consistent and broad improvements across a wide range of downstream tasks. For example, beyond brain captioning, our model outperforms all baselines on 33 out of 38 evaluation metrics reported in Table 2.
>
> **C4**
>
> The ablation study is not sufficiently convincing. There is no explanation of the task being experimented on, and the authors should provide the final results, not just the loss.
>
> **R4**
>
> We appreciate the reviewer’s feedback and apologize for the lack of clarity regarding the experimental setup. The ablation study is conducted across all tasks during brain instruction tuning on subject 1, and the reported loss is the average loss across these tasks. We will update the details about settings in the revised version.
>
> Our decision to present loss curves was intentional: As pointed out in lines 405–411, left column, the comparison of the convergence speeds of the orange and green lines helps illustrate the impact of both region and positional encodings. (We do not plot metrics during training at each step since generation is time-consuming)
>
> However, we agree that final performance metrics are essential for a complete picture. We have now computed these metrics for most downstream tasks, and the results are shown below. The results are consistent with our original claim based on the loss curves. Note that it also validates our response to the reviewer's concern 1.
>
> [Table](https://anonymous.4open.science/api/repo/MindLLM-FAA4/file/rebuttal/ablation.png)
>
> We will include the final results of all tasks in the revised version.

---

> > ### Comment · Reviewer_Gvse · 2025-04-08
> >
> > While I remain skeptical about the theoretical basis for using voxel positions in attention computation (which seems experimentally motivated), I acknowledge the authors' efforts in addressing some concerns. Accordingly, I have adjusted my score from 1 to 2.

---

> > > ### Author Response · Authors · 2025-04-09
> > >
> > > We thank the reviewer for their feedback and for reconsidering their score.
> > >
> > > Given that the reviewer still remains skeptical about the theoretical basis for using voxel positions alone, we would like to politely further clarify that our design choices are not merely *experimentally* motivated, but are motivated by both neuroscientific intuition and established literature, and ultimately validated by empirical results (Section 4.6).
> > >
> > > Specifically, as noted in lines 124–126 (right column), it is generally accepted that the human brain functions broadly exhibit spatial consistency across subjects. For example, the motion of the right body is mainly related to the left hemisphere, while the right hemisphere handles most of the left body's movement [1].
> > >
> > > Furthermore, prior work has shown a coupling between a voxel's cognitive function and the anatomical role of the voxel in MRI, which is to some extent reflected by its spatial location in the brain [2, 3]. Hence, we theoretically infer that voxel positions are related to brain function characterization.
> > >
> > > Therefore, we argue that our design aligns well with the unique properties of fMRI data, in contrast to images or text. The empirical findings further support this modeling choice.
> > >
> > > Finally, to address potential variability across subjects (e.g., anatomical and functional shifts), we introduce **region encodings** (lines 147, right column – 203, left column), which provide additional neuroscientific grounding and act as a calibration mechanism to complement the raw positional information.
> > >
> > > [1] McManus, I. C. (2002). Right hand, left hand: The origins of asymmetry in brains, bodies, atoms, and cultures. Harvard University Press.
> > >
> > > [2] Zhang, X., Liang, C., Wang, N., Wang, Y., Gao, Y., Sui, C., ... & Wen, H. (2023). Abnormal whole-brain voxelwise structure-function coupling and its association with cognitive dysfunction in patients with different cerebral small vessel disease burdens. *Frontiers in Aging Neuroscience*, *15*, 1148738.
> > >
> > > [3] Liu, C., Jing, J., Jiang, J., Wen, W., Zhu, W., Li, Z., ... & Wang, Y. (2024). Relationships between brain structure-function coupling in normal aging and cognition: A cross-ethnicity population-based study. *NeuroImage*, *299*, 120847.

---

### Decision · Program_Chairs · 2025-05-01

**Decision:**

Accept (poster)

**Comment:**

This is a paper presenting a cross-subject question answering model that analysis fMRI patterns and answers questions about what was being processed or intended at a given time. The authors perform modeling at a voxel / parcel level use a neuroscience informed attention process where they replace the attention key with the location in the brain (specific regions have more or less specific functions, at least we know that neighboring voxels have similar functions). The authors also introduce a brain instruction. The authors show that their model has good performance relevant to baseline, and the more interesting part is that it is agnostic to subject identity and that it performs better using combinations of data across subjects (the authors are requested to add these results to the paper). The authors are also requested to add all their answers to the comments in the paper as well as the other results and changes they promised.